# Speech-induced suppression during natural dialogues
Joaquin E. Gonzalez [1] ✉, Nicolás Nieto [2,3], Pablo Brusco [4], Agustín Gravano [5,6,7] &
Juan E. Kamienkowski[1,4,8]

When engaged in a conversation, one receives auditory information from the other's speech but also from their own speech. However, this information is processed differently by an effect called Speech-Induced Suppression. Here, we studied brain representation of acoustic properties of speech in natural unscripted dialogues, using electroencephalography (EEG) and high-quality speech recordings from both participants. Using encoding techniques, we were able to reproduce a broad range of previous findings on listening to another's speech, and achieving even better performances when predicting EEG signal in this complex scenario. Furthermore, we found no response when listening to oneself, using different acoustic features (spectrogram, envelope, etc.) and frequency bands, evidencing a strong effect of SIS. The present work shows that this mechanism is present, and even stronger, during natural dialogues. Moreover, the methodology presented here opens the possibility of a deeper understanding of the related mechanisms in a wider range of contexts.

Motor-induced suppression is a ubiquitous mechanism in the brain where the response to self-produced stimuli is suppressed or attenuated by means of an efference copy. As its name suggests, this mechanism refers to a duplicate of a motor command sent to the sensory pathway, that contains a subtractive signal for canceling a predictable sensory input caused by their own behavior. This mechanism is the key to filtering out internally generated signals from the external stimuli when interacting with the environment, from self-made tickles to eye movements[1–4]. In the field of speech processing, this effect is known as Speech-Induced-Suppression (SIS), and it has been found that the suppression of self-produced speech is very specific to the stimulus expected by the brain, and not due to global inhibition of the auditory cortex activity[5–8]. Thus, speakers are able to quickly adjust the produced speech to match the expected stimulus. Also, based on this mechanism, they can avoid getting confused by their own voice, as is the case of auditory hallucinations in some pathologies like schizophrenia[9]. This effect has been shown with single-unit recordings in monkeys and humans[10,11], and in EEG/MEG studies, but typically finding an N1/M100 attenuation, and using single syllables as stimulus[5–8,12].

The neuroanatomical functioning of the acoustic aspects of continuous speech perception has been, and remains, difficult to characterize[13,14]. Using mostly controlled stimuli, where the participants were passive listeners, it has been shown that speech perception and processing occur mainly in the left temporal and frontal lateral hemisphere, whereas the representation of acoustic stimuli is bilateral[15,16]. Moreover, several studies have shown that speech comprehension regions are closely related to speech production[14,17–20]. They suggest that, in adverse or noisy situations, left fronto-temporal regions have a greater interaction with the primary auditory cortex. Furthermore, a synchronization between the brain activity and the acoustic features of the speech signal was observed, called cortical entrainment, which was shown to play an important role in the intelligibility of the information present in speech[14,17,21–23].

Although these analyses have led to a better understanding of speech processing, the set of neural and cognitive systems involved in traditional perceptual tasks, performed in the laboratory with controlled stimuli, just partially overlaps with the systems involved in natural speech perception and comprehension[16]. This makes the representation of natural speech

¹Laboratorio de Inteligencia Artificial Aplicada, Instituto de Ciencias de la Computación (Universidad de Buenos Aires - Consejo Nacional de Investigaciones Cientificas y Tecnicas), Buenos Aires, Argentina. ²Instituto de Investigación en Señales, Sistemas e Inteligencia Computacional, sinc(i) (Universidad Nacional del Litoral - Consejo Nacional de Investigaciones Cientificas y Tecnicas), Santa Fe, Argentina. ³Instituto de Matemática Aplicada del Litoral, IMAL-UNL/CONICET, Santa Fe, Argentina. ⁴Departamento de Computación, Facultad de Ciencias Exactas y Naturales, Universidad de Buenos Aires, Buenos Aires, Argentina. ⁵Laboratorio de Inteligencia Artificial, Universidad Torcuato Di Tella, Buenos Aires, Argentina. ⁶Escuela de Negocios, Universidad Torcuato Di Tella, Buenos Aires, Argentina. ⁷Consejo Nacional de Investigaciones Científicas y Técnicas, Buenos Aires, Argentina. ⁸Maestria de Explotación de Datos y Descubrimiento del Conocimiento, Facultad de Ciencias Exactas y Naturales - Facultad de Ingenieria, Universidad de Buenos Aires, Buenos Aires, Argentina.
✉e-mail: joaquin.gonzalez6693@gmail.com

difficult to study through traditional methods. Different authors are currently working on the analysis of natural stimuli[21–24], but still in partially constrained scenarios. The realization of more naturalistic, unconstrained tasks brings new challenges. Firstly, when using electroencephalography (EEG) signals, there is an increased contamination with muscular and ocular artifacts. Secondly, there is also an increase in the complexity of the stimuli and the number of features that can have an impact on brain activity[23,24]. Lastly, the ongoing nature of the interaction with the environment or another person in everyday tasks, such as in a dialog, involves an overlap of different perceptual, motor and cognitive signals. Consequently, analyzing the brain representation of continuous auditory stimuli would require defining the onset of the embedded events, such as the speech act, to perform a traditional Event-Related Potential (ERP) approach.

To tackle these problems several authors have used *encoding models*[21–23,25,26], which allow experimenters to explore the relationship between brain activity and the different features of one continuous stimulus. The first step of these models consists of a feature extraction stage from the stimuli presented to the participant. When using auditory stimuli, the extracted features are usually the envelope of the audio signal, the spectrogram, and the speaker's voice pitch, among others. Also, by filtering the brain activity in different EEG frequency bands, it is possible to quantify and compare the impact and representation of the stimuli features on each typical EEG band[27–29]. In the second step, typically a linear regression model is trained by fitting its parameters to find the relation between the extracted features and the brain's activity. The parameters or weights of the model, are an estimate of the multivariate Temporal Response Functions (mTRF)[25,26], and express the relationship between the stimuli and the brain activity, representing the response of the EEG signal to the stimulus[21,25–29]. Then, the trained model can be used to predict EEG signals from unseen speech stimuli[21,23,27–29]. Importantly, the interpretation of the mTRF is similar to that of an ERP, the major differences being the assumption of a linear relationship between stimulus features and EEG signals, and the fact that the mTRF isolates the response to specific features entangled within a stimulus. However, to interpret the mTRF in such a way, it is necessary to evaluate the model's predictive power[21,25,26,28–31].

In the present work, we perform a novel experiment in which participants are involved in natural, unscripted dialog, and analyzed the brain's response to either speaker's voice separately. We use encoding models to provide a methodology to untangle the brain activity related to different features of either speaker's speech, even if they occur simultaneously. We hypothesize that encoding models will have a better performance when used in natural scenarios and that the response to self-produced speech would be specifically suppressed during dialog, whereas the response to the other participant's voice would not, even if both are speaking at the same time.

## Results

### Brain representation of acoustic features

To study speech processing during a natural dialog, we conducted an experiment in which we simultaneously recorded high-density EEG from two participants while performing several trials of the Objects game of the UBA games corpus[32–34]. This is a cooperative task designed for dialog studies, where the participants had to communicate through speech to place an object in a specific position on the screen. Each participant also had a directional microphone to collect high-quality audio synchronized with the EEG (Fig. 1a, see also Supplementary Fig. 1). From this dataset, we realized two different studies. First, we implemented an encoding model to analyze the representation of acoustic features from natural speech stimuli in the listener's brain, replicating and extending results from previous works. We interpret higher correlation values as markers of higher representation in the brain, as discussed in Section "Discussion". To this end, we used the time intervals where only one of the participants was speaking, for which the dialog intervals were categorized into four conditions: Silence, Only speaker 1, Only speaker 2, and Both speaking. Secondly, having validated the model's functioning, we performed the analysis of SIS over different dialog conditions.

An encoding model was trained for each participant, using the audio features extracted from its partner's audio as input to predict each individual EEG sample using the stimulus features extracted from the previous 600 ms (Fig. 1b). This procedure was applied to each electrode and EEG frequency band separately. The correlation between the predicted and the recorded EEG signal was used as a measure of the model's performance (see Section "Encoding models" for a detailed explanation of the model fit and the results presented).

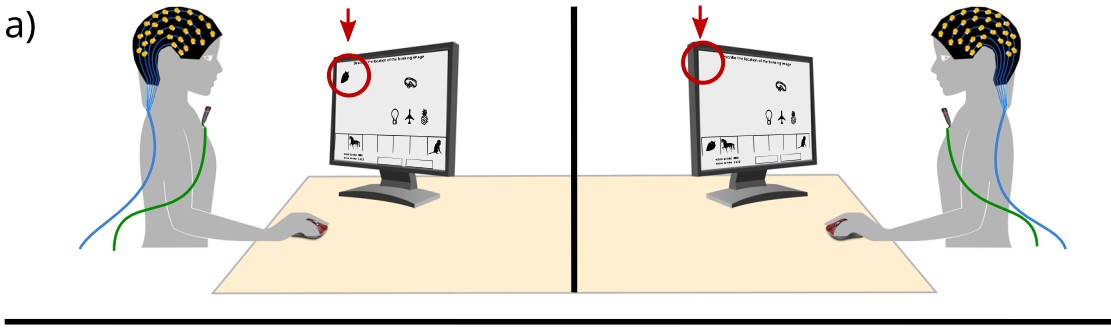

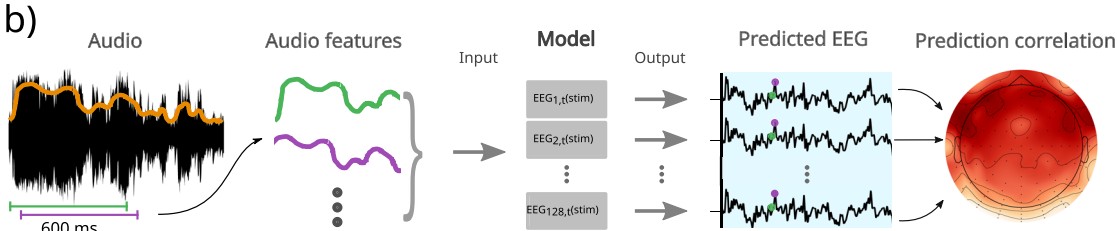

**Fig. 1 | Experimental design and analysis pipeline. a** The participants were sitting facing each other, with an opaque curtain that prevented visual communication. They were presented with a screen with objects distributed on it, and a task that required verbal communication. Uttered speech by each participant was recorded using a microphone, and the brain activity was recorded using 128 EEG channels. **b** Audio signals from which the audio features were extracted (envelope orange) and used to first train an encoding model through Ridge regression fitted to the EEG of each channel. Then, the features were used as input to predict the EEG of those same channels. Pearson correlation between the predicted and the recorded EEG signal is used as a measure of the model's performance.

The first step was to assess and endorse the functionality of our model, by replicating previous results but in a more natural and dynamic environment (Figs. 2 and 3, Supplementary Note 2 and Supplementary Figs. 2 and 3)[21–23,35]. We mainly focused on the spectrogram[21,23,36,37], but also used the envelope, both of which are among the most widely used speech features[21–23,26,38]. We obtained a mTRF waveform similar to previous work and reproduced previous results on the lateralization effect in the correlation values, and even slightly improve the performance of previous models using naturalistic stimuli (Figs. 2 and 3). We obtained an average correlation value in the Theta band of 0.26 for the envelope and 0.37 for the spectrogram (reaching 0.28 and 0.41 in frontal electrodes), where the previous average correlation values were approximately 0.26 and 0.09 respectively[23]. The results of the envelope and the remaining acoustic features (pitch and shimmer) can be found in Supplementary Fig. 5.

The correlation coefficients obtained from the spectrogram model averaged across subjects reached 0.41 in the Theta band and 0.36 in the Delta band (Fig. 2a). In the Theta band, the spectrogram presents a better predictive power in the frontal lateral regions of both hemispheres, more lateralized to the left, where language-related areas are found (Fig. 2b; Wilcoxon signed-rank: $z = 0$, $n = 12$ (d.f.: 11), $p$-value = 0.00048). This is in accordance with the literature that situated most of the speech and language-related regions lateralized to the left[39–49]. In contrast, the envelope shows similar values in left and right hemispheres, supporting previous works that found that the envelope of an artificial audio signal is lateralized to the right[35] (Fig. 2a). It is possible to observe a lateralization effect in all frequency bands (Supplementary Fig. 4). Importantly, before exploring the mTRFs, we assess that the obtained results were statistically significant and that the model is robust by comparing its results with 3000 random permutations of the preceding stimulus features. This procedure was repeated for each electrode and participant separately, using 5-fold cross-validation to avoid data-

partition-related misleads. The number of subjects for which the obtained correlations were significantly different from the random distribution in all folds was ($13.3 \pm 1.4$) subjects for Delta, ($16.2 \pm 1.0$) for Theta, ($14.3 \pm 1.5$) for Alpha, ($9.2 \pm 1.8$) for Low Beta and ($5.8 \pm 1.9$) for the broad-band (0.1–40 Hz) (see the first row of Supplementary Fig. 5b, and Supplementary Fig. 3.1c for spatial distribution). The statistical significance of these results shows that the model is robust, and reliably expresses the relationship between stimuli and brain activity, hence the mTRFs can be interpreted as the response in the EEG to the continuous stimuli[21,26,29].

The mTRFs from the audio spectrogram in the Theta band presented multiple peaks that resemble the auditory evoked potentials[50,51] (Fig. 3a). Previous work with encoding models and natural speech had reported a similar pattern with latencies approximately 50 ms lower[21,26]. This difference in timing could be explained by the Causal filters used in this work to preserve the causality in the EEG signal samples, which could have an impact on the mTRF fitting and time response. This had been previously addressed in ref. 22, where they found that the causal filters preserved the overall shape in the mTRF, but introduced a time delay of about 50 ms, which is consistent with the delays in our results. For more details on the Causal filters used, and a comparison with the results from Non-Causal filters, please see Supplementary Note 5, Supplementary Figs. 7 and 8 and Tables 1 and 2. We repeated this analysis by re-referencing the EEG signal to the average of all electrodes to gain detail on the differences between electrodes and the polarization in the scalp. The results can be found in Supplementary Note 6 and Supplementary Fig. 9. Moreover, within the spectrogram, the more important features are the audio mel-bands ranging from 583 to 2281 Hz, which correspond to the ones where human speech carries more information. This was assessed by a Threshold-Free Cluster Enhancement test on which those frequency bands presented the most significant results (Fig. 3b, c).

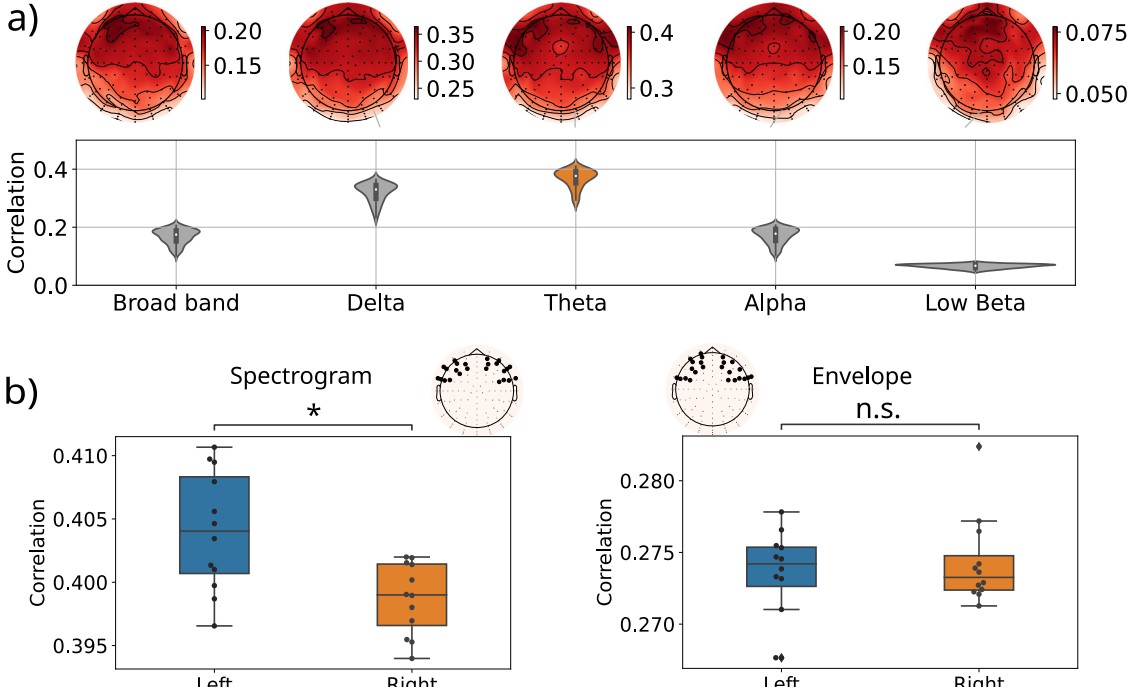

**Fig. 2 | Model performance while listening to external speech. a** Correlation coefficients for every frequency band and their spatial distribution obtained from the spectrogram model. The correlation values were obtained for each electrode and averaged across the 5 folds within each participant, then averaged across participants. The top panel A shows the spatial distribution of the averaged correlation values, to better determine the regions where higher correlation is achieved. The distribution shown in the lower panel consists of those same values but presented in a violin plot, for an easier comparison across frequency bands. **b** Correlation

distribution for left and right electrodes indicated in the topographic figure, for the models using spectrogram and envelope as input features. The electrodes were chosen as the 12 presenting higher correlation values in the frontal region for each hemisphere and a signed-rank Wilcoxon test was performed to compare the values obtained in each hemisphere ($N = 12$ independent samples). The correlation values for the spectrogram show a significant lateralization effect towards the left hemisphere, with a $p$-value of ~0.0005, whereas the envelope shows no significant difference ($p$-value ~ 0.38). Significance: n.s $p$-value > 0.05, *$p$-value < 0.001.

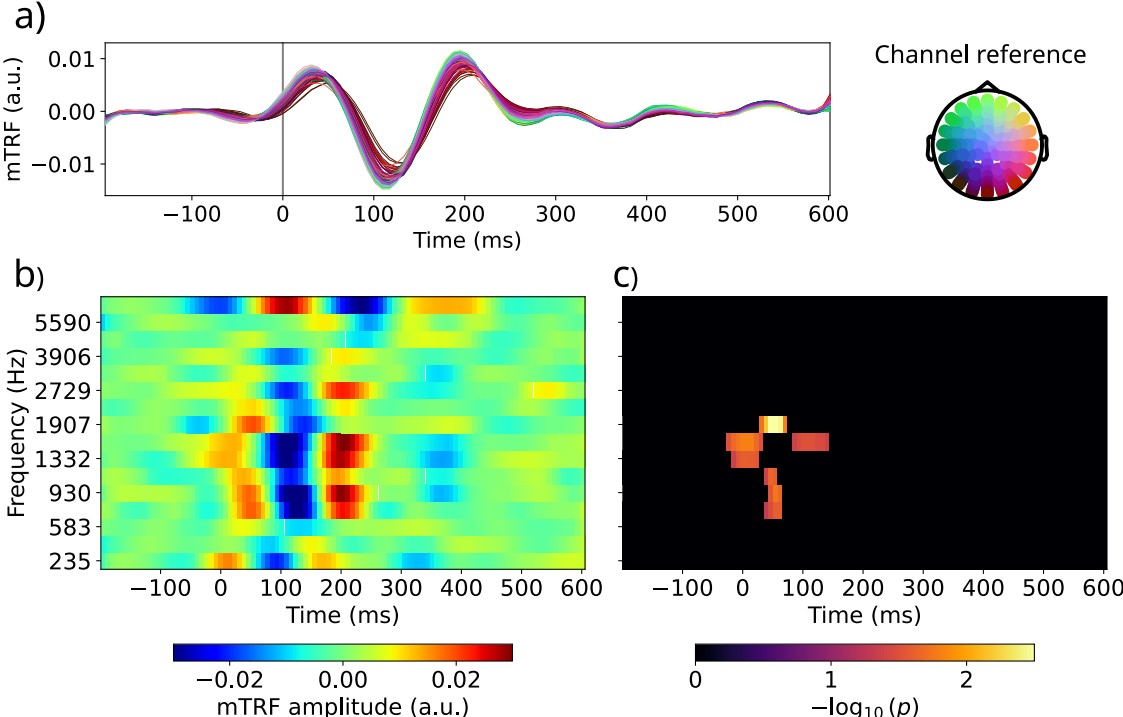

**Fig. 3 | Theta band mTRFs to the audio spectrogram while listening to external speech.** Panel **a** shows the mTRF for each electrode, averaged across participants and mel-bands. The position of each electrode is indicated by the scalp plot on the right. Panel **b** shows the mTRF to each of the spectrogram features (each mel-frequency band) averaging the responses over electrodes. Panel **c** shows the *p*-values in negative logarithmic scale from a TFCE test applied to the mel-frequency band mTRF of all subjects separately (*N* = 18 independent samples, d.f.: 17). The mTRFs represent the response in the EEG signal to each time lag of the audio features. The time axis represents the time elapsed between the audio feature being pronounced and the instant in the EEG signal being predicted. For representation purposes, pre-stim time lags are included in the figure, but the predictions were made only from positive times, to avoid providing the model with information from future time-points. Please see Supplementary Note 4 and Supplementary Fig. 6 for a detailed explanation of the time axis on these figures.

## Speech-induced suppression

During dialog "self-produced" speech is also present as an available auditory stimulus but, as in many other sensory modalities, its information should be available beforehand by means of the efference copy, in order to discriminate from external stimuli. We confirm this hypothesis in natural dialogs by analyzing the impact of the spectrogram on the EEG signal in different dialog stages across frequency bands, but focus on the Theta band, as this is the band with a significantly higher predictive power.

The results presented in Fig. 4 show the correlation coefficients for each frequency band and the spectrogram mTRFs from the Theta band for all conditions. Figure 4a shows the mTRFs to the spectrogram feature when listening to external speech (condition (E)), and the correlation coefficients of all frequency bands (both already presented in Figs. 2 and 3). This response was significantly different from background noise (Silence) as supported by a pairwise comparison and Cohen's d-prime computation (Fig. 5a; (E) vs Silence: uncorrected *p*-value < $10^{-5}$ for all channels, mean Cohen's d' = 3.22). Moreover, the Bayes Factors comparison H0 (no difference between conditions) and H1 (difference between conditions) presents decisive evidence in favor of H1 in all channels (Fig. 5a; $log_{10}(BF_{10}) > 2$).

The response to the external speech (E) was also significantly different from the one to self-produced speech condition (S) as supported by pairwise comparison, Cohen's d-prime computation, and $BF_{10}$ (Fig. 5a; (E) vs (S): uncorrected *p*-values < $10^{-4}$ for all channels, mean Cohen's d' = 3.22; $log_{10}(BF_{10}) > 2$). Moreover, the mTRF in the (S) condition presents no evidence for response in the Theta band as it is clear from Fig. 4b. Its response is similar to the background noise, i.e. when both participants are silent (Silence) (Fig. 4b, e). This was supported by a pairwise comparison and Cohen's d-prime computation between the correlation coefficients from (S) with Silence conditions for the Theta band (Fig. 5a; uncorrected *p*-value > 0.12 for all channels, mean uncorrected *p*-values = 0.52, mean

Cohen's d' = 0.31). Interestingly, the evidence is consistent with the H0 hypothesis of no difference ($log_{10}(BF_{10}) < 0$), although it is not enough to support it (Fig. 5a).

These results were replicated when both participants were speaking. In this case, we only kept segments longer than 600 ms in which both participants were speaking and estimated the mTRF with the signal of the other's microphone (E|B) or with one's own (S|B) (Fig. 4c, d). Figure 5b shows that the (E|B) condition presented a significantly larger response than the (S|B) and Silence conditions (uncorrected *p*-values < 0.05/128 = $3.9 \times 10^{-4}$ for 84 channels; mean uncorrected *p*-values < $4.4 \times 10^{-4}$), which was supported by Cohen's d-prime (mean Cohen's d' > 1.16) and the BF10 (BF10 > 1.12 for all channels in both comparisons, mean BF10 > 2 in both comparisons). Moreover, (S|B) did not present significant differences with Silence (uncorrected *p*-value > $5.3 \times 10^{-4}$ for all channels, mean uncorrected p-values = 0.22) and a very low d-prime (mean Cohen's d' = 0.56). The Bayes Factor analysis did not present conclusive results in favor of H0 or H1 (mean BF10 = 0.15).

It is worth mentioning that, even though the same patterns of results arose from the mTRFs in the isolated speech and both speaking conditions, there were some differences related to methodological considerations. Firstly, when comparing the (E) and (E|B) conditions the amplitude of the mTRF in the latter condition is significantly reduced (Fig. 5c; (E) vs (E|B): uncorrected *p*-values < $10^{-4}$ for all channels, mean Cohen's d' = 1.22). This is primarily attributed to the model being trained with fewer samples, as indicated in Supplementary Note 7 and Supplementary Fig. 10. Secondly, when comparing the (S|B) condition with Silence there was some evidence of response in fronto-lateral regions, which is also present in the comparison between (S|B) and (S). This could be explained by the fact that, even though the microphones were directional and there was a curtain between the participants, some attenuated signal of the other's speech could be present in the signal of one's own microphone. Thus, in the condition in which both

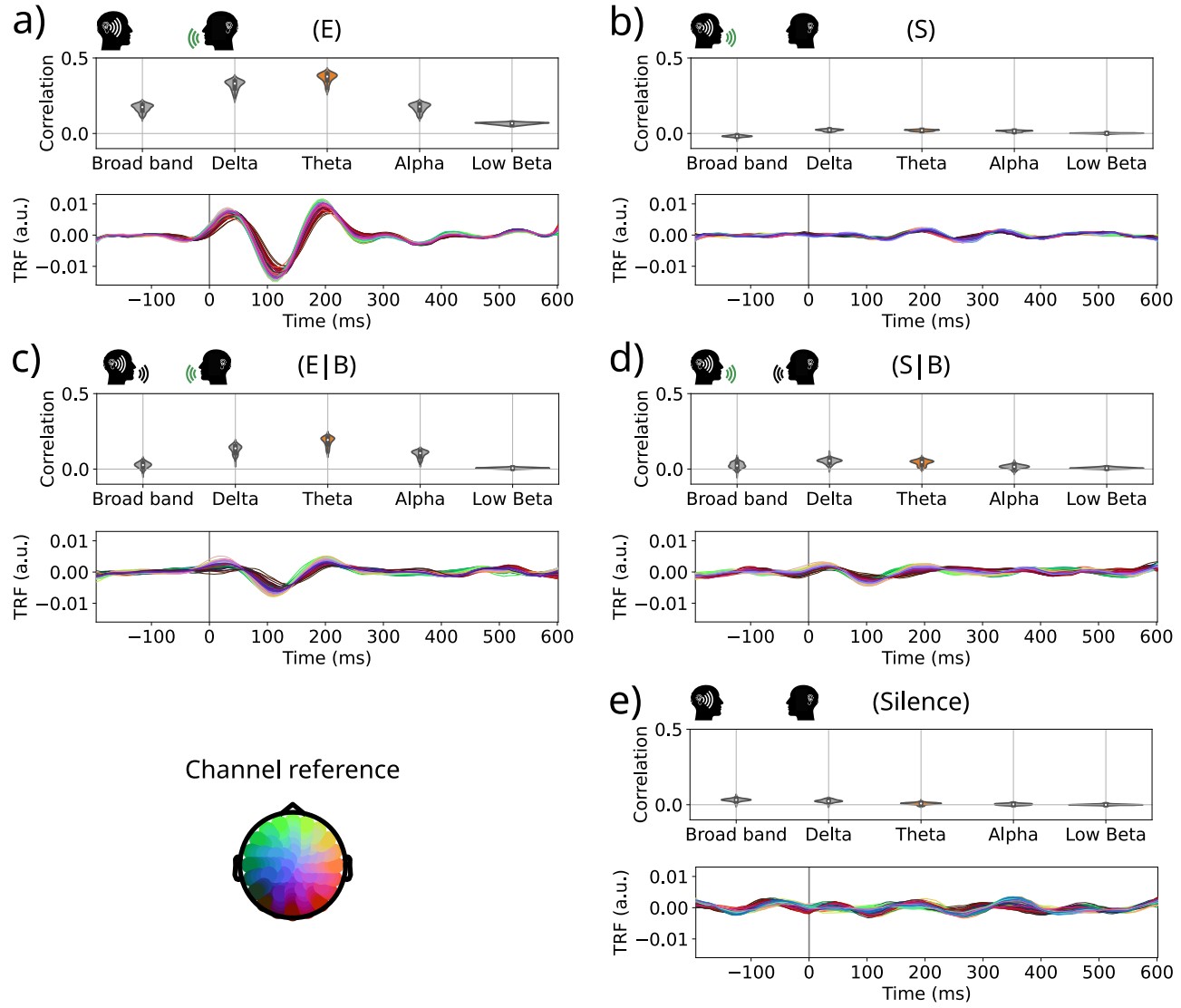

**Fig. 4 | Speech-Induced Suppression: Correlation values of all frequency bands and Theta band mTRFs to the spectrogram for every dialog condition. a** Listening to external speech (E); **b** Listening to self-produced speech (S); **c** Listening to the external speech while both are speaking (E|B); **d** Listening to the self-produced speech while both are speaking (S|B); **e** Silence. Mean number of samples per participant: NA/NB = 49,034 (range between [17,825–78,259]), NC/ND = 2692 ([1207–4617]), NE = 31,586 ([15,207–73,464]). Again, pre-stim time lags are included in the figure for representation purposes, but the predictions were made only from positive times, to avoid providing the model with information from future time-points.

were speaking at the same time, the EEG is capturing that signal in its own microphone.

In all, the self produced audio signal brings no or minimal information to predict the EEG signal. Hence, these results suggest that there is a Speech-Induced-Suppression effect blocking the information from one's own voice.

The results presented here are also supported by the phase-locking value (PLV) between the EEG signal and the envelope of the audio signal at different lags. To this end, the phase synchronization between the EEG signal of each electrode in the Theta band, and the envelope signal band-pass filtered between 4–8 Hz was computed. The analysis presented in Fig. 6a, c shows that both situations of (E) and (E|B) present a higher synchronization at a latency of ~125 ms, where the mTRFs also reached their maximum absolute value (Fig. 4a, c). Furthermore, the largest values of the (E) condition are distributed over the same frontal lateral regions where the encoding model found the strongest representation of the spectrogram feature (Fig. 2a). Conversely, the remaining dialog conditions (S, S|B, Silence) presented much lower PLVs at all time-lags, and no clear synchronization or regions of impact at any time (Fig. 6b, d, e).

This analysis not only determines the most synchronous channels to the envelope at a certain time lag, but it can also specifically measure the latency at which the overall synchronization is maximum. This could bring new information concerning the delays in the processing of auditory stimuli. Furthermore, it replicates the SIS results in a model-free approach based on the phase of the signals rather than the amplitude (as the encoding models) reassuring the consistency of the result.

## Discussion

In this work, we studied the neural representation of acoustic features from natural and unrestricted dialogs between pairs of participants. Using encoding models, we estimate the response in the EEG signal to different audio features from the perceived stimuli used for the analysis, obtaining similar responses to the evoked potentials known to discrete stimuli with established onset times from previous works. In accordance with our hypothesis, we found that encoding models achieve a good performance when the participants are engaged in natural dialog, where the prosody is known to carry an important part of the information, and also, where the participant needs to engage in the dialog to fulfill a task[52,53]. Furthermore, we

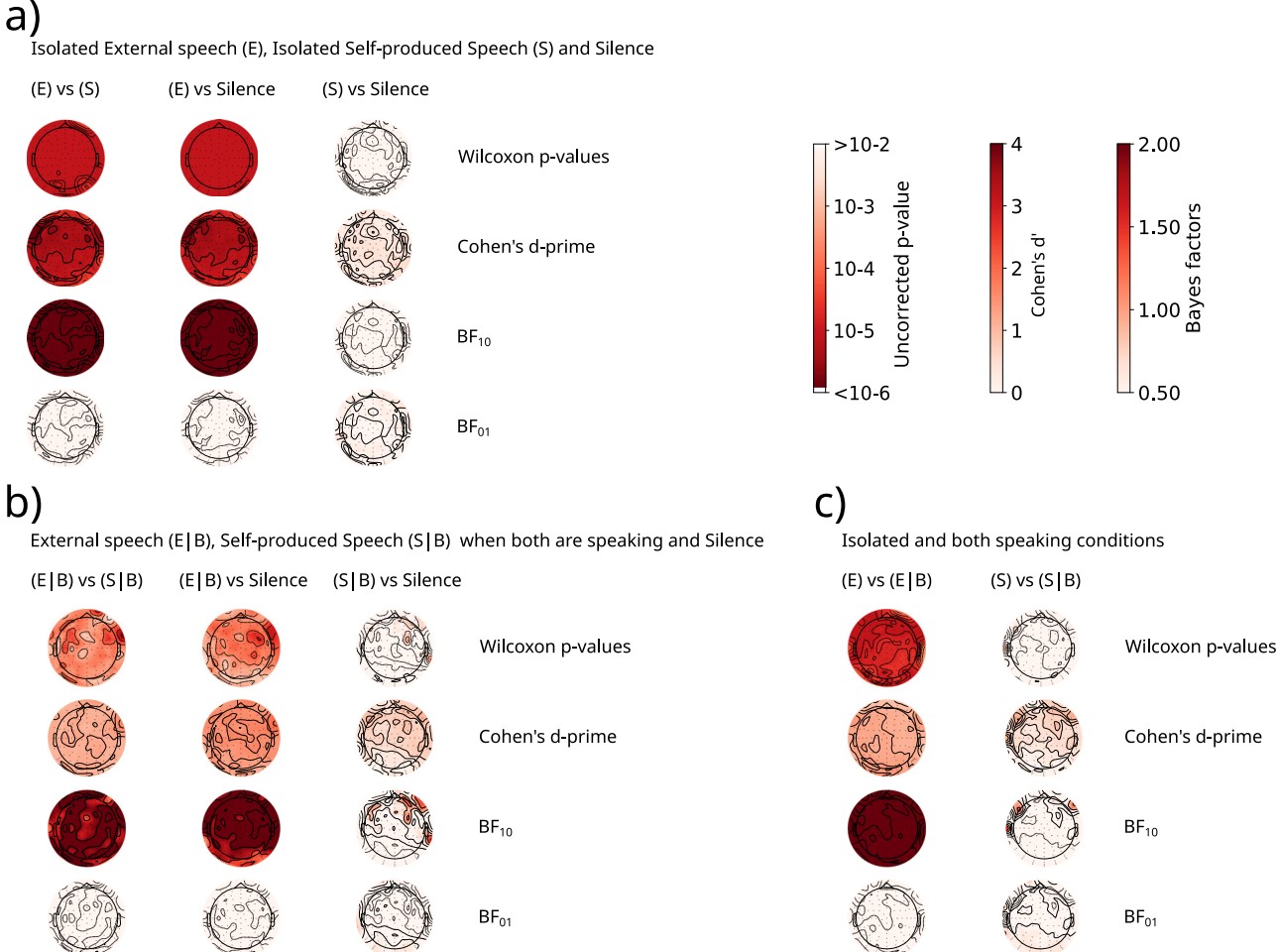

**Fig. 5 | Comparisons between the average correlation values of each electrode in the Theta band from different listening conditions: Results from Wilcoxon signed-rank test, Cohen's d-prime, and Bayes Factors in favor of the hypothesis H1 ($BF_{10}$), and in favor of the hypothesis H0 ($BF_{01}$) ($N$ = 18, d.f.: 17).**
**a** Comparison between isolated External speech, Self-produced speech and Silence. **b** Comparison between External speech and Self-produced speech when both participants are speaking, and Silence. **c** Comparison between isolated and both

participants speaking conditions. The conditions are abbreviated as follows: Listening to external speech (E), Listening to self-produced speech (S), Listening to the external speech while both are speaking (E|B), Listening to the self-produced speech while both are speaking (S|B). Uncorrected p-values should be compared with a threshold of $0.05/128 = 3.9 \times 10^{-4}$ (Bonferroni corrected-threshold), also see Supplementary Note 8 Supplementary Fig. 11 for False-Discovery Rate (FDR) corrected $p$-values.

observed a total suppression of the brain response to self-produced speech, even in situations where both participants were simultaneously speaking.

Previous studies using EEG in humans also measure the performance as the correlation between the predicted and the original EEG signal[21,23]. Considering that the mTRF are interpreted as an evoked response to the stimulus feature used as input in the model[26,28–31], we propose that the regions with better predictive power in the model, would correspond to those with a higher representation of the stimulus feature, always considering first the statistical significance of the correlation values of the predictions. These studies used several audio features as input for the model, in particular the envelope and the spectrogram. For the envelope they obtained a mean correlation value of around 0.03 using Audiobooks[21], 0.26 using pre-recorded audio from a sentence database (TIMIT)[23,54], and 0.07 using Movie Trailers (MT)[23]. For the spectrogram, all the obtained values were under 0.1 (Audiobooks: 0.06, TIMIT: 0.09, MT: 0.02). Moreover, when combining these features as input for a model, the mean correlation values slightly improved the performance in TIMIT, reaching 0.35 and 0.1 for MT. In the case of natural dialogs, we obtained better performance than the three cases for both features (spectrogram: 0.41, and envelope: 0.23). These results were also similar to studies with ECoG that use spectrotemporal features ($r_{onset}$ = 0.26 and $r_{sustained}$ = 0.34)[36] or the envelope ($r2 \simeq 0.19$, r = 0.43)[38] from TIMIT, MT and modified audios. It is important to consider that we

only compared our results with the uncorrected correlation values reported in ref. 23 and refs. 21,23 also presented noise-ceiling correction correlation values[55,56] but such correction relies on trials and it is not possible to be computed in continuous unscripted data.

Two possible explanations for the better correlation coefficients are that the predictions in this work are computed for each frequency band separately, whereas the main results reported from the previous studies used a broader band (1–15 Hz). Focusing on the Theta band could increase the correlation values as the representation of these auditory features seems to take place in a narrow frequency band, as was the case in ref. 21 (2.5 increase when using only Theta). Also, the adaptation of the methodology for automatically computing the optimal alpha parameter of the ridge regression, which could contribute to the better results obtained in the test set, as it is intended to reduce the overfitting to the train set (see Supplementary Note 9 and Supplementary Fig. 12 for more details).

It is worth mentioning that in our case, as in the case of Audiobooks[21], we obtained a better performance with the spectrogram model than with the envelope. We hypothesize that this is because the spectrogram presents more detailed information about the stimulus. Those other studies also explore features like phonemes or the onset of an utterance that yield better results. It would be interesting to explore those other features in our dataset in the future.

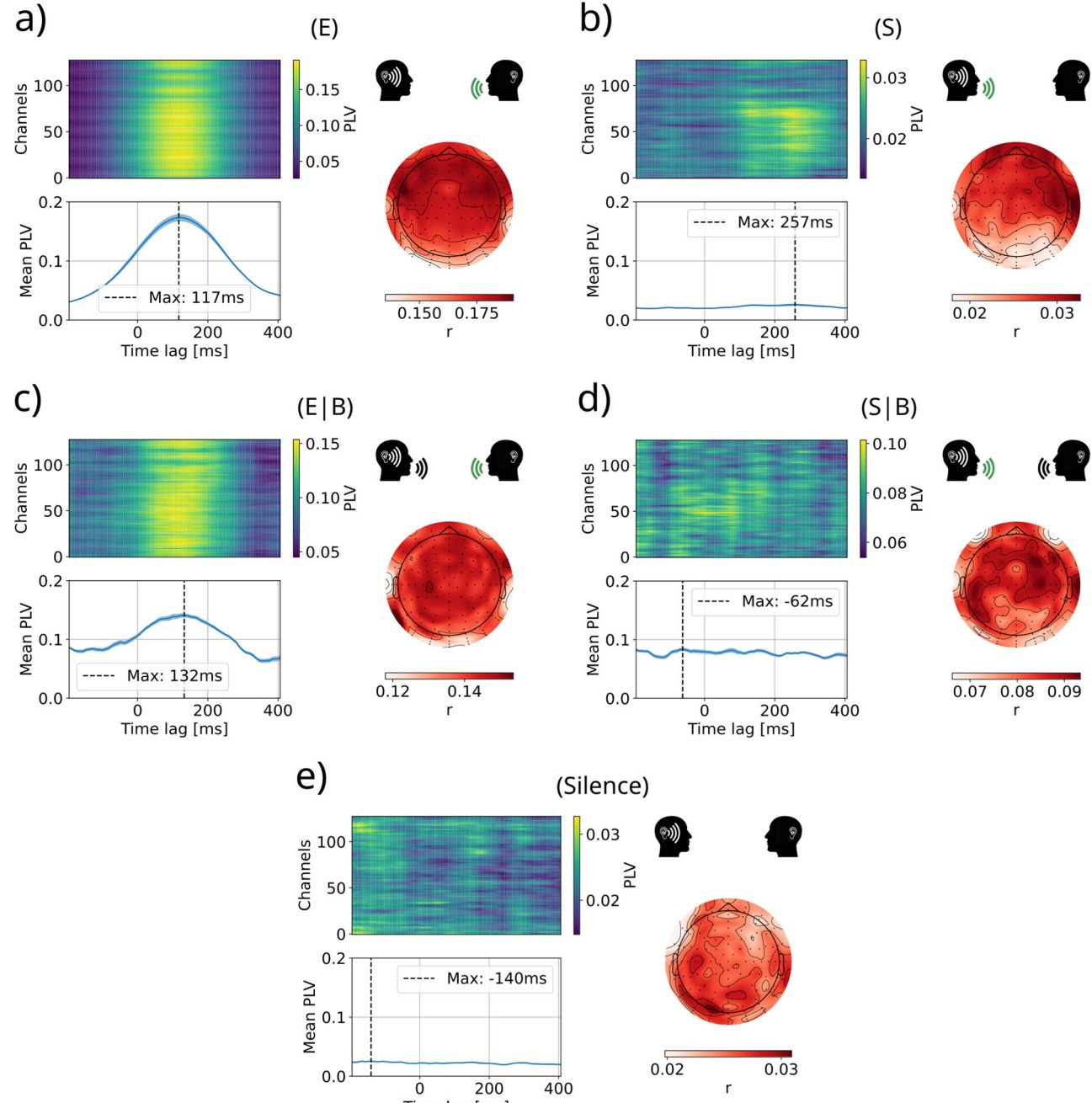

**Fig. 6 | Phase-locking value (PLV) between the EEG signal of each electrode and the envelope signal, averaged across participants for all dialog conditions.** **a** Listening to external speech (E); **b** Listening to self-produced speech (S); **c** Listening to the external speech while both are speaking (E|B); **d** Listening to the self-produced speech while both are speaking (S|B); **e** Silence. Each panel shows the phase synchronization between each EEG channel and the envelope feature (top left), and the average values and standard deviation across channels (bottom left), for all time-lags between −200 ms and 400 ms. The time lag 0 corresponds to the EEG and envelope from matching instants, negative latencies indicate that the EEG signal precedes the auditory signal (making it impossible to have a causal effect), while positive lags indicate that the brain activity follows the auditory signal. On the right side of every panel, the topographic distribution of phase-locking values for the time lag of maximum average synchronization.

In general, the encoding framework allows us to also explore the temporal, spatial and frequency distributions of the brain representations of the different features. Overall, the obtained results were similar to those in previous studies that focused on specific aspects of brain responses, but in this case, reproduced all in one dataset, strengthening the consistency and robustness of these findings. Consistent with previous studies using different analytical methods[22,50,51,57], and taking into account the delay introduced by causal filters, the largest peaks in the TRFs and the highest synchronization values were found at approximately first 200 ms. In particular, Etard et al.[22]

used decoding models to find the time lag that better predicted the envelope from the EEG signal and Perez[57] used Gaussian Copula Mutual Information (GCMI) synchronization analysis. These time-windows are compatible with the classical auditory N1 obtained with ERP analysis in similar studies[50,51].

In terms of frequency analysis, our results presenting higher predictive power in the Theta band agree with previous work[21,22], on which they used encoding and decoding frameworks respectively. In both cases, they looked into the performance of the models for different frequency bands and found

Theta (in the first case) and Delta and Theta (in the second, being Delta slightly better) to be the frequency bands with higher correlation values. Here, we showed that the effect is statistically significant and that it is also present in natural speech and in Spanish. The saliency of the Theta band is also consistent with other studies that refer the synchronization of the brain activity and the acoustic features of the speech signal as the cortical entrainment effect[17,58]. This effects was explored in English and Mandarin Chinese[59], but all languages have speech rates in the same range (Mandarin: 5.18 ± 0.15, English: 6.19 ± 0.16, and Spanish: 7.82 ± 0.16)[60].

Regarding the localization, Lalor and colleagues obtained the larger responses in fronto-centro-temporal electrodes[25,26]. Similarly, Di Liberto et al. focused their analysis on lateralized fronto-central electrodes (approximately FC3-6, C3-6) after exploring the whole scalp[21]. Those regions overlap with the region with higher representation found in this work (approximately F3-8, FT7-8, C3-5). The envelope was mainly represented in the frontal region of both hemispheres, with a slight, yet non-significant, lateralization to the right, in line with previous work[35]. Conversely, the spectrogram was represented mainly in the lateral frontal electrodes of the left hemisphere, which could indicate that the spectrogram feature encodes not only acoustic information, but also language-related information. This is consistent with previous findings on passive listening situations showing an interaction of Broca's area with the primary auditory cortex[14,17–20].

Speech-induced suppression was previously studied by analyzing the modulation of evoked potentials, mainly the N1/M100, in different scenarios[5–9]. These studies analyzed the difference in brain response to alien and self-produced speech in the traditional "Talk-Listen" paradigm, using mainly recorded speech from pronounced vowels and introducing noise, pure tones, frequency shifts, or delays to the perceived stimuli[5–8,12]. Their results mostly showed partial attenuation in the response to the intact self-produced speech, but not when the produced speech was altered before feedback. Only in a few cases, does this attenuation reach full suppression of the measured response both in EEG and ECoG[7,8,61]. Their findings presented evidence indicating that the SIS mechanism is very specific, and it only applies when the stimuli exactly match an expected outcome of the produced speech. The encoding approach allows us to deal with continuous unscripted speech, even in the context of many simultaneous sources, in particular, to differentiate a response to self-produced or external stimuli when both are produced at the same time. In this context, two speakers are engaged in a conversation and the impact of self-generated speech and external stimuli can be naturally assessed. In this context, we noted a more pronounced inhibition of self-generated speech compared to the majority of earlier investigations. A potential rationale for this phenomenon might stem from the inherent alignment of encoding models with the continuous stream of speech, while the ERP method predominantly captures transient responses aligned to the onset of a stimulus (often ill-defined). Hence, the strong attenuation of the response to self-produced speech sustained over time suggests that the suppression effect might increase after the attenuated response to the self-produced stimuli onset (reported in previous studies[5,6,8]). This effect could take place by profiting from the mechanisms known to parse speech information in the brain, in which different regions of the auditory cortex present onset or sustained responses to speech stimuli[36].

In parallel, recent work studying phase synchronicity had found a significant and instantaneous synchronization between the EEG signal and the self-produced audio envelope, and a 100 ms lagged synchronization between the EEG signal and the audio envelope of a speaker[57]. However, the authors suggest that the immediate synchronization could be due to muscular artifacts in the EEG that correlate with the produced speech. The strong suppression of the response and synchronization to the self-produced speech features observed here could indicate that previous results were indeed caused by correlation to muscular artifacts. The results presented here using the PLV method applied to external stimuli support the latency results obtained from the encoding analysis through the mTRFs. Moreover, the spatial distribution of the synchronization presents its higher values in the same electrodes where the encoding model found the maximum correlation, even when both methods present different nature in their formulation, one analyzing the amplitude of the signals (encoding) and the other the phase synchronization (PLV).

Even though the precision in time is important for the SIS effect, some studies also explore the network that supports the monitoring of the own speech with fMRI[62] or combining EEG and MRI[8]. They showed that activity along the superior temporal sulcus and superior temporal gyrus bilaterally was greater when the auditory stimulus did not match the predicted outcome. These regions generally related to speech perception, can be also linked to a control system that predicts the sensory outcome of speech and processes an error signal in speech-sensitive regions when there is a mismatch. These results are in accordance with N1/M100 sources[8], which is similar to the spatial distribution of the brain representation of speech that is suppressed found in this work.

The SIS effect is proposed to be generated by the efference copy of the expected output of the speaker's own voice, sent to the primary auditory cortex (Helsch's area) to cancel such signal from the perceived stimulus[61]. Although this mechanism is not yet fully comprehended, the approach proposed here could serve to confront it with an alternative hypothesis such as a selective attention mechanism similar to the one proposed in discriminating speakers within a crowd (cocktail-party problem)[63,64]. Moreover, it could open the way to explore how specific it is in terms of the different features of speech and to study it in more natural scenarios and with more natural stimuli, other than isolated vowels.

It is important to take into consideration some methodological decisions and their possible implications to compare the results with previous findings. One potentially confounding factor is related to the muscular artifacts related to speech articulation and the application of Independent Components Analysis (ICA). During the dialogs, we asked participants to minimize unnecessary movements to avoid upsetting the electrodes, but we did not restrict movements during the task. Also, they wore an EEG cap, had to remain seated, and were separated by an opaque blanket hanging between them (to discourage facial and hand gestures). During preprocessing, we aimed to remove as much unrelated noise as possible without affecting neural signals. We use ICA mainly to remove eye movements and blinks, discontinuities that correspond to bad electrodes or single electrodes with high noise for an interval of time, and some extreme muscular artifacts probably related to jaw and neck. First, we used two semi-automatic criteria for eye movements and blinks: EyeCatch[65] and ADJUST[66], and discontinuities[66]; keeping ICs that showed a peak in alpha frequencies. It is important to note that the same components were used for the whole EEG recording, independently of whether the participant was speaking or listening. Thus, we expect that no correlations with the spectrogram or envelope for these components were introduced by bias in component selection. Second, we also removed other muscular artifacts. We identified these components as having sharp spatial distributions located in the edgemost temporal electrodes (over the ears), with high-frequency spectra (typically flat or U-shaped spectra), and without a peak in alpha. These components are usually just referred to as 'muscle components' in the bibliography[67–69] and tutorials on artifact removal with ICA (for instance[70]). They more likely capture neck or jaw movements, but not lips or tongue. The lips and tongue movement certainly could have a spectrum more concentrated in lower frequencies, than in an occipital-frontal dipole[71]. Furthermore, over the different articulators, the jaw is probably the one with the lowest correlations with the speech spectrogram[72]. Thus, the ICA preprocess was not expected to remove much of the spectrogram responses and thus introduced a potential bias towards lower responses in the speaking condition. On the contrary, as we did not remove components specifically related to tongue or lips artifacts, in the worst case, it could be expected some immediate increase in the response in that condition, as discussed by Pérez et al.[57]. As mentioned before, it is important to note that all the artifact removal was performed in the continuous data, before the separation into conditions.

Another note of caution refers to the overinterpretation of the latencies. As shown in the Supplementary Note 5 (Supplementary Fig. 7) and consistent with[57] the latencies depend on the filter choice. Here, we showed that this choice did not affect the performance of the model and the spatial distribution, and to some extent also the waveform of the responses. We based our choice on the criteria that the future must not influence the EEG prediction, i.e. that the amplitude of the TRF in the baseline must be close to zero, and thus we use zero-phase non-causal filters. Still, other choices are possible if the latencies were of special interest (see ref. 21).

This work is part of a novel trend in the field of neuroscience, advocated in understanding brain function in natural scenarios[24]. Here we work with an unconstrained experimental task not explored to date, under the hypothesis that constraint tasks allow to describe the building blocks but not how they interact between them, hence their interaction in a dynamical, unconstrained environment remains unexplored. To the best of our knowledge, this is the first time that an effect of speech-induced suppression is shown in a natural and dynamic situation as dialog, even when listening while speaking, by taking advantage of both the unique experimental setup and the analytical methods presented here. The results in this work show a clear and robust way to measure such effects, supported by the replication and extension of most of the previous findings studying the representation of acoustic features of continuous speech in the brain; all this with a trial-free analysis methodology. In terms of the representation of acoustic features, the evidence of activity in the language regions extended the previous results that suggested its interaction with the primary auditory cortex in passive listening to more natural scenarios[14]. The performance of the models motivates the realization of such more "natural" experiments of listening while producing speech with devices with more precise localization, such as magnetoencephalogram (MEG) or electrocorticogram (ECoG), to untangle the brain mechanisms of the SIS.

## Methods

### Participants

The data used in this work consists of 10 sessions, in which 20 subjects, 10 women and 10 men aged between 19 and 43 years (M = 26.4, SD = 6.3) and Spanish native speakers participated. Only one session was later discarded due to the poor quality of the EEG signal in one of the subjects[73]. All participants gave written informed consent and were naive about the aims of the experiment. All the experiments described in this paper were reviewed and approved by the ethics committee of the Centre of Medical Education and Clinical Research "Norberto Quirno" (CEMIC) (Protocol 435), qualified by the Department of Health and Human Services (HHS, USA). All ethical regulations relevant to human research participants were followed.

### Experimental design

In each session, two participants were sitting facing each other in a recording booth, separated by an opaque curtain that prevented visual communication. Each participant had a computer, where a graphical interface allowed them to develop a joint task that required verbal and unrestricted communication[32,33]. During each session, between 17 to 30 trials (average of 24.3 trials) of 1–5 min were carried out with an average of 82.6 s (SD = 61s), for a total of 5.3 h of recording. A diagram of the experimental design is shown in Fig. 1a.

### EEG recording and preprocessing

EEG activity was recorded using two BioSemi Active-Two systems at 128 positions each with a sampling rate of 1024 Hz. The electrooculogram (EOG) and linked mastoid reference were also recorded. Preprocessing was performed in MatLab using the EEGLAB toolbox[74]. The original filter on raw data was performed using a Finite Impulse Response (FIR) filter (pop_eegfiltnew)[75] in three steps: First a high-pass filter (low cut-off = 0.1; order = 16,896), then a low-pass filter (high cut-off = 100; order = 100), and a Notch filter (cutoff = [49, 51]; order = 3380). The required filter order/transition bandwidth is estimated as the transition bandwidth is 25% of the lower edge. These filters were all zero-phase non-causal filters, applied

following the instructions of the EEGLAB toolbox, where standard procedure for applying band-pass filters is the successive filtering of high-pass and low-pass causal filters[76].

The intervals between trials were eliminated from the recordings and independent component analysis (ICA) was applied to the remaining data[77], with the aim of removing ocular and muscle artifacts. From the 128 components generated by the Infomax algorithm, artifactual components were identified, first, using EyeCatch[65] and ADJUST[66] plugins for EEGLAB. As suggested by the developers, the selection was supervised by an expert (one of the authors). Components presenting a peak in spectra in the alpha band were unmarked. Then, the same expert (one of the authors) marked muscular components based on the following criteria: 1. spatial distributions with sharp peaks located in the edgemost temporal electrodes (over the ears), 2. with high frequency spectra (typically flat or U-shaped spectra), and 3. without a peak in the alpha frequency band. Overall, an average of 22 out of 128 components were removed per participant. EyeCatch and ADJUST usually agreed in the eye movement components but ADJUST also provided the discontinuities (isolated bad electrodes or bad intervals in one electrode), a total of 17.5 components were discarded using these methods. Finally, 4.5 components were identified on average using the criteria for muscle artifacts.

After the preprocessing, carried out in the acquisition stage, the EEG signal was filtered in the following frequency bands of special interest for EEG analysis, in order to discern the effects in each one: Delta (1–4 Hz), Theta (4–8 Hz), Alpha (8–13 Hz), Low Beta (13–19 Hz), and a broad ERP band (0.1–40 Hz). A minimum-phase causal FIR filter was applied, using the implementation in the MNE Python library[78], which calls Scipy's signal firwin function, to implement an FIR filter using the window method[79]. The parameters were set to use a 'hamming' window, to pad the edges with the signal edge values, and to introduce a 'minimum' phase lag in the causal filter. The transition zones were automatically determined to minimize the artifacts introduced by the filtering. For more details of the filtering process and the differences between Causal and Non Causal filters, please see Supplementary Note 5 and Supplementary Tables 1 and 2. Finally, the Z-scores of the signal were calculated and the sampling frequency was reduced to 128 Hz by sub-sampling.

Speech from each participant was recorded on separate channels with a TASCAM DR-100 digital recorder (44.1 kHz, 16 bits) and using two Rode HS-1 speech microphones mounted on the participant's head. The recorded audio signal was downsampled to 16 kHz and synchronized with the EEG signal. This procedure was performed using a low-resolution copy of the audio recording that was incorporated to the EEG recordings as an analog input. With this signal, the time shift that maximizes the cross-correlation between the two audio copies was obtained and the signals were synchronized (see Supplementary Note 1).

### Feature extraction

The main feature used in this analysis is the mel-spectrogram computed from the audio signal. This was computed using librosa[80] in Python, which uses the fast Fourier transform (fft) to decompose a discrete signal into the frequency space. The signal was decomposed into 16 mel-bands of up to 8 kHz. This frequency decomposition is thought to reflect the filtering performed by the human auditory system[81], and has been used extensively in previous works in the field[36,37,82]. The computation of the fft and the spectrogram was performed on windows of 125 non-overlapping audio samples, in order to obtain a signal with the same sampling frequency as the EEG signal. In this way, the audio samples of the spectrogram at each instant do not contain information corresponding to the signal at future points, which could lead to a misinterpretation of the model adjustment times. The speech envelope was calculated using scipy's 1.7.1[79] implementation of the Hilbert transform on the audio signal, then taking the absolute value of the real and imaginary components[21,23,38]. To reduce the temporal resolution, the average was taken in non-overlapping windows of 125 audio samples, resulting in a signal with a sampling frequency of 128 Hz. Finally, the signal was normalized between [0,1] before using it as input in the model. For a detailed

description of the shimmer and pitch extraction please see Supplementary Note 3.

### Interval selection

For the proposed analysis, we classify the dialog status in each sample according to who was speaking or listening. First, we define an Inter-Pausal Unit (IPU) as a maximal speech segment from a single speaker that is surrounded by pauses longer than 100 ms. IPUs in our corpus were manually aligned to the audio signal by trained annotators[34]. Thus, we obtained the beginning and ending times of all uttered sentences by each speaker, and we can define intervals where only one participant is speaking, both are speaking, or there is silence.

### Encoding models

A model was fitted for each participant to each EEG channel independently (Fig. 1). Each sample corresponded to an interval where the participant had been uninterruptedly listening to their partner speak for at least 0.6 s (condition E in our manuscript). Using overlapped sliding windows with 1 time-point step, all the valid intervals within each session were extracted (for each condition separately). On average, the number of samples ($N_{samples}$) used for the analysis was around 2692 (range between [1207–4617]) per participant for the condition of both participants simultaneously speaking and 49,034 (range between [17,825–78,259]) per participant for exclusive listening situations. For each EEG sample, the previous 600 ms of the audio feature of interest (temporal delays) were taken as model inputs based on previous results[21,36,37]. The first 3 ms were discarded to exclude approximately the time it takes for the sound wave to reach the speaker located 1 m away. Since the sample rate of both the audio and the EEG signals were 128 Hz, the 600 ms segments resulted in 77 time-points (or delays; $N_{times}$). These vectors were used as model input matrix with a final dimension of $[N_{samples} \times N_{times}]$. In the case of the spectrogram, as there are 16 frequency bands, the dimensions of the vectors became $[N_{times} \times N_{features}]$ corresponding to a $77 \times 16 = 1232$ vector.

Encoding models use linear regressions to find the relationship between the audio features and the neural activity. Many times, due to the collinearity of the input features, and also to prevent overfitting, a Ridge regression is used[21–23,36,37]. Ridge regression counts with a regularization parameter which reduces the effect of collinearity and helps prevent overfitting by penalizing higher weight values. For a detailed description of the Ridge parameter selection see Supplementary Note 9.

The encoding models were fitted for each electrode and participant separately, over a 5-fold cross validation procedure that intended to avoid getting spurious results due to poor data partitioning. The mTRFs were averaged over the 5 folds first, yielding one result per electrode per participant. Finally, these were averaged across participants to obtain the results for the Grand Average of subjects, presented in the results section.

We assess the performance of the model as the Pearson correlation between the predicted EEG, reconstructed for all samples in the test set, and the actual EEG signal. Similar to mTRFs, these correlation values were calculated for each channel and participant separately, then averaged across the 5 folds within each participant and finally averaged across participants, obtaining one value for each electrode. The results were presented as spatial distributions of the mean values across participants, or mean values across participants and channels. To summarize the results in the main text, we reported: 1. the average value across all electrodes, and 2. the highest correlation value across electrodes. This maximum value is usually reported because it is not expected that the speech signal impacts all the electrodes in the same way, and thus averaging correlation values across 128 electrodes could largely shadow the results.

A recent study[23] also presented the correlation values from the predictions with a normalization-correction[55,56], along with the uncorrected correlation values. The noise-ceiling correction normalizes the performance of the model with the maximum theoretical performance ($C_{max}$) given by the data itself. However, it relies on trial repetition for the computation of the $C_{max}$ making it not possible to implement in continuous unscripted data.

This correction is useful to distinguish a bad performance due to a poor model or to noisy data, yet it exacerbates correlation values. It would be interesting to develop a similar measure for continuous unscripted data.

### Statistical analysis

**Model's significance**. To endorse the results obtained from the model, in terms of latencies, frequency bands and impact regions, the statistical significance of the model predictions needs to be assessed. Then, the mTRFs reliably represent the response to the specific feature of continuous stimuli. To do this, a permutation test was performed, where 3000 surrogate models were trained with 3000 different random permutations of the input matrix for each participant[83]. The permutations were realized over the samples axis, keeping the temporal structure of each sample and its correlation structure across the 600 ms (see Supplementary Note 10 and Supplementary Fig. 13 for more details). The surrogate models were then used to make predictions on the original evaluation set. The alpha hyper-parameter used was the one obtained for the original model for each case (feature, frequency band, and participant). Then, 3000 correlation values were obtained from the random predictions, from which a null distribution was generated in order to compare it with the correlation value obtained by the model using the original data. The p-value of the original model prediction is:

$$p - value = \frac{N_{\rho_{Rand} > \rho_{True}} + 1}{N_{perm} + 1} \tag{1}$$

Where $N_{\rho_{Rand} > \rho_{True}}$ corresponds to the number of times that the correlation of a random permutation of stimuli was greater than that obtained with the original data, and $N_{perm}$ indicates the total number of permutations performed (3000).

This process was repeated for each electrode in each fold of an unshuffled 5-fold cross-validation procedure, in order to avoid spurious results due to poor data partitioning. The significance threshold was set at 0.05 and corrected by Bonferroni (0.05/128), conservatively considering that the test will be performed on the 128 channels separately. Only channels that passed the test (p-value <0.0004) in all folds were considered significant. Finally, the results of each participant, indicating which electrodes were significant, were summed over participants, yielding a scalp plot with the number of significant subjects for each electrode. See Supplementary Fig. 5c.

**Lateralization statistical test**. The comparison between correlation values from frontal lateral electrodes was performed with a signed-rank Wilcoxon test using statannot, the original version of statannotations library for python[84]. 12 electrodes of each hemisphere were used, giving 12 samples ($n = 12$, d.f.:11).

**Threshold-Free Cluster Enhancement test**. In order to identify the significant time-points and frequency bands for the prediction of the EEG signal across participant, the mTRF fitted for the spectrogram feature was subject to a 1-sample permutations test (Threshold-Free Cluster Enhancement, TFCE) across participants ($N = 18$, d.f.: 17) as implemented in the MNE python library, using 4096 permutations and a threshold parameter starting in 0, with a step of 0.2[85]. The resulting p-values were masked when values exceeded the 0.05 threshold, and presented in logarithmic scale for visualization purposes.

**Models comparison**. To compare the results of models trained in different dialog conditions, a signed-rank Wilcoxon test was performed on the correlation values of all participants for each channel separately. This was done with the Wilcoxon function from *scipy.stast* module version 1.7.1[79]. The encoding models were fitted for each participant individually, obtaining 18 correlation values for each electrode (after averaging over the 5 folds). This was repeated for the 5 dialog conditions, obtaining a $5 \times 18$ matrix with the correlation values for each electrode. Paired comparisons between every combination of conditions were performed

using a signed-rank Wilcoxon test ($N = 18$) for each electrode, yielding a scalp distribution of $p$-values. Given that each hypothesis was tested in the 128 channels separately, the significance threshold was corrected by Bonferroni for multiple comparisons (but see also the results with the FDR correction below). To estimate the evidence in favor of both the null and the alternative hypothesis (H0 and H1) we estimated the evidence in favor of one hypothesis over the other using the Bayes Factors[86,87]. As before, the null hypothesis (H0) implies that the model performances in both condition are equal, and the alternative hypothesis (H1) implies that there is an effect (a difference between conditions). We presented results on both the BF10, which the evidence in favor of H1 over H0, and the BF01, which is the inverse of BF10. Roughly speaking, as we used logarithmic scale values larger than 0 correspond to positive evidence in favor to one of the hypothesis (H1 in the case of BF10), values larger than 0.5 correspond to substantial evidence, and values larger than 2 correspond to decisive cases. Finally, Cohen's d-prime was also computed on the same data, to assess the effect size between them in standard deviation units.

## Phase-locking value

The phase synchronization between the envelope of the audio signal filtered between 4–8 Hz and the EEG signal from the Theta band was computed by the following equation[88]:

$$\left| PLV(\tau) \right| = \left| \frac{1}{n} \sum_{t=1}^{n} \exp(i(\theta_{env}(t) - \theta_{eeg}(t - \tau))) \right| \qquad (2)$$

Where $\theta^{env}$ corresponds to the phase angle of the envelope signal, $\theta^{eeg}$ to the phase angle of the EEG signal for one channel, extracted by means of the Hilbert transform, and $n$ is the total length of both signals. This method averages unit vectors in the complex plane, where the phase of those vectors corresponds to the phase difference between signals at each time ($t$) (i.e. all the samples that belonged to the corresponding dialog condition were used as if they conformed one unique trial, from which the phase synchronization was calculated by averaging the complex unit vectors of phase differences between the EEG and envelope signals). Then, taking the absolute value yields a PLV between 0 and 1, where higher values correspond to more synchronous signals. The synchronization values between the audio envelope and every channel of the EEG were computed separately for each participant, and then averaged across participants. This analysis was repeated time-shifting the envelope signal with respect to the EEG signal, allowing to find the time-lag when the synchronization between signals is maximum. $\tau$ represents the time lags between the signals, ranging from $-200$ to 400 ms in a similar manner as ref. 57. Negative latencies indicate that the EEG signal precedes the auditory signal (making it impossible to have a causal effect), while positive lags indicate that the brain activity follows the auditory signal. Zero lag is when the auditory and EEG signals are synchronous.

## Reporting summary

Further information on research design is available in the Nature Portfolio Reporting Summary linked to this article.

## Data availability

Speech data was part of the UBA Games Corpus, which was already released in an institutional public repository (https://ri.conicet.gov.ar/handle/11336/191235)[73]. All data (EEG, audio and transcriptions) was deposited into figshare and can be accessed at the following URL: https://doi.org/10.6084/m9.figshare.22647313.

## Code availability

The code to replicate the results obtained in this manuscript can be found in https://github.com/jegonza66/SIS-during-natural-dialog/.

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

## Acknowledgements
We thank Mauro Veneziano for his contribution on the initial versions of the analysis and Victoria Peterson for her thoughtful early contributions to the project. The authors were supported by the National Science and Technology Research Council (CONICET), the University of Buenos Aires and Universidad Torcuato Di Tella. The research was supported by the University of Buenos Aires (20020190100054BA), The National Agency of Promotion of Science and Technology (PICT 2018-2699) and the National Scientific and Technical Research Council (PIP 11220150100787CO).

## Author contributions
J.E.K. and A.G. designed the experiment and collected the data. J.E.K. performed the pre-analysis of the EEG data. P.B. and A.G. performed the pre-analysis and annotation of the speech data. J.E.G., N.N. and J.E.K. analyzed the data, wrote the first version and revised the manuscript. P.B. and A.G. revised the manuscript.

## Competing interests
The authors declare no competing interests.
