## [Peer Review File · Communications Biology]

Reviewers' comments:

Reviewer #1 (Remarks to the Author):

This is a well conceived study on the difference in neural responses to speech of others vs our own speech. The introduction provides a good background discussion and motivation. The main finding appears consistent with prior literature showing reduced EEG responses to our own speech. The results, if confirmed, would generalize artificial laboratory tasks to natural dialog. Unfortunately the study has a small sample size (N=9 pairs of speakers/listeners) and the main result – the contrast in neural responses while both individuals are speaking – is uncertain, because statistics are not adequately performed. Generally, the manuscript is lacking in presentation of the statistics and analysis methods. Perhaps this can be fixed with a thorough rewrite of the Results section, but the small sample size will remain a hurdle.

In the following are comments to text in the order it appears in the manuscript.

“We were able to reproduce previous results and even slightly improve the performance of previous models using naturalistic stimuli (Fig. 2.2). We obtained a maximum correlation value of 0.57 for the envelope and 0.67 for the spectrogram, where the best previous correlation values were approximately 0.5 for the envelope [23].” The values reported in Fig. 2.2 are indeed high. However, the presentation of the results is lacking. Neither caption nor text describe the results adequately. This is true for most of the Results section. Fortunately all this has an easy fix. Just say what the results are: No where in the figures can one see values 0.57 or 0.67. Instead, it appears from panel B that values max out below 0.41 and 0.28. Besides, reporting maxima is not particularly insightful as maximum values are inherently noisy. Panel A shows distributions, but it is not said over what these distributions are taken (electrodes? Subjects?). If it is over subjects, was the mean or maximum taken across electrodes for each subject? It is also unclear what the different EEG bands mean. Are you modeling the power of those signals, or the filtered signals itself? The methods only say that you filtered in those bands.

From reading the rest of the paper, I am guessing you are not attempting to predict power in those bands, but instead the filtered signal itself in the band. That is a fine idea, but there is no need to estimate new TRF for each of these filtered signals. TRFs can capture all frequencies in an input→output mapping. There is no need to filter the output in different frequency bands. Typically the way you do this is to find a single mapping to the broadband signal. To measure how well each frequency band was predicted you can just measure the coherence spectrum between predicted output and real output, i.e. eeg. This is quite conventional linear systems. The only caveat is that regularization has a nonlinear effect on the estimate. So different bands may be variably affected, and so the TRF (vs the decomposed TRFs) may yield somewhat different results. But it does reduce the number of free parameters (by a factor of 5, to be exact, as you are using 5 different versions of the same signal in your current approach, broadband + 4 pass-band filtered version). Given the small sample size of this study, this may be helpful.

Panel 2.2 C top shows a scalp plot that is not explained. The time courses are not clear. There are multiple curves, are these for multiple electrodes or a single electrode and multiple subjects? Or perhaps each curve is for one of the different spectrogram features? The caption of figure 2.3 seems to suggest that. It should really be made clear already in figure 2.2. They look all very similar, maybe they can all use the same TRF and thus have the same number of parameters with the envelope, so the correlation coefficients are comparable. As it is, the higher values with the spectrogram (Figure 2.2B) could simply result from the larger number of free parameters. How exactly were the different bands in the spectrogram combined?

"In the Theta band, the representation of the spectrogram is stronger in the frontal lateral regions of both hemispheres, more lateralized to the left, where language-related areas such as Broca are found (Fig. 2.2 B; Wilcoxon test: $z=12$, $n=12$, p -value = 0.034)." How was this set of electrodes selected? The p -value here is borderline, if more than one set of electrodes was tested, this clearly is no longer meaningful. Also, the word "representation" is something you can propose in the discussion, but it's not the data. What you probably mean to say is that the spectrogram predicts those electrodes best, or that the correlation is highest for those electrodes. It does not mean that this part of the brain "represents" the spectrogram. Indeed, low level features like that are likely best "represented" in the auditory cortex at a much faster timescale. Why the theta (power?) is more robustly driven by the spectrogram features in those frontal electrodes is worth a discussion, in the Discussion section.

"It is possible to observe a gradual lateralization effect from frontal electrodes in higher frequency bands to more lateral in the Theta band (Fig. 2.2 A)." I am sorry, but I can't see that in figure 2.2A. And even if there was a subjective impression on those figures, colormaps can be incredibly deceptive. I would like to see some sort of statistical analysis to support this observation. Otherwise it's just tea leafing.

"The number of subjects for which the obtained correlations were significantly different from the random distribution in all folds was ..." In the methods this test was performed separately per electrode. But here there is only talk of subjects. Do you mean where at least one electrode generated significant correlation? Or how else are these numbers established? Incidentally, there are only 9 pairs of subjects. The numbers here suggest that this was done for 18 subjects. Is that right? Also, it is not clear if this analysis is for the spectrogram features or the envelope.

"non-altered evaluation set" I don't know what you mean by "altered". You mean "altered"? What was altered in the rest of the data, but no on the "evaluation set"? And in the "random permutation" what exactly was permuted? Details here matter. As with all random stats, if correlation structure in the data is perturbed (except for the one specifically to be tested) the Null distribution can be quite incorrect.

"Moreover, within the spectrogram, the more important features are the audio mel-bands ranging from to 1907 Hz, which correspond to the ones where human speech carries more information." How was this established? If Figure 2.2C shows TRF to different frequency bands, they all look very similar. You would need some sort of statistical test to say one is bigger than another, perhaps do stats across subjects.

"The mTRFs present an absence of response in any frequency band in both situations, and are similar to the response to background noise when both participants are silent (Fig. 2.3 E). This was confirmed by a pairwise comparison between the correlation coefficients from (S) and (S|B) conditions with Silence." The lack of evidence for an effect is not the same as evidence for a lack of an effect. But that is not even what is interesting in this experiment. Suppression does not need to be so strong as to not have a response at all to self speech. Additionally, poor correlation with simultaneous speaking may be due to noise in the EEG during speaking, which is substantial – something you acknowledge later in the result section.

Rather, what you want to show is that the response is weaker, i.e. a contrast between panels C and D. Namely, both are speaking, but the EEG response to the other speaker is stronger than the EEG response to self speech. This analysis you do in fact perform a little later. It is the central finding of the paper, as it controls for speaking noise in EEG, and shows an effect, rather than arguing for a lack of an effect, which is always difficult to establish. And this begin the

"Fig. 2.4; S vs Silence: uncorrected p-value > 0.12 for all channels, mean uncorrected p-values = 0.52; (S|B) vs Silence: uncorrected p-value > 5.3×10^{-4} for all channels, mean uncorrected p-values = 0.22" If you really want to show evidence in support of the Null hypothesis, you should report Bayes factors.

"(Fig. 2.4; uncorrected p-values < 3.9×10^{-4} for 84 channels; mean uncorrected p-values < 4.4×10^{-4})" It is never clear in this paper how statistical comparisons have been done. Are correlation values computed per subject and then the stats performed across subjects? The degrees of freedom or the sample size needs to be reported for each statistical evaluation in the results section. It's hard to imagine small p-values with N=18 subjects, and Bonferroni correction across 128 electrodes (as stated in the methods).

Also, comparisons rarely tell the reader what measures are being used? I am guessing mostly correlation with theta band EEG predicted from spectrogram? Or from an envelope?

Also, it is never explained if TRFs are computed per subject, or together for all subjects.

Now to the actual result shown in figure 2.4 "This was statistically confirmed by pairwise comparison between the correlation values from (E|B) and (S|B) when both participants were speaking simultaneously, ... (Fig. 2.4; uncorrected p-values < 3.9×10^{-4} for 84 channels)." To me, this is the central result of the paper, so it deserves some attention. The relevant scalp plot in figure 2.4 looks rather noisy in the sense that neighboring electrodes have quite different p-values. Real brain activity is expected to have smoother spatial distribution due to the blurring function of CSF and scalp. P-values also tend to be noisy, certainly on N=18. My first suggestion is to do a proper multiple comparison correction, like FDR. That 84 of 128 channels are below some arbitrary hand-picked p-value is not appropriate statistical comparison or reporting. The second recommendation is to show the effect size, rather than p-values, for instance, Cohen's d-prime. That will be a lot more informative, and likely smoother in space. My guess is that with N=18 and 128 electrodes, you are not adequately powered to detect statistically significant differences.

"This method averages single tone vectors in the complex plane, where the phase of those vectors corresponds to the phase difference between signals at each time (t). ... This analysis was repeated time-shifting the envelope signal ... to find the time-lag when the synchronization between signals is maximum."

PLV -- as introduced in Lachaux 1999 (reference below) -- averages phase angle across trials. It is not clear what was done here. What are "single tone vectors"? And over what are averages computed, i.e. what exactly is t in equation (2)? Given the argument about time-shift, it seems the average was taken across time samples within utterances? Multiplying the angle difference with t makes little sense to me. Also, how are angles computed? Usually Hilbert transforms of some band-passed signal. But here the stimulus (envelope?) does not seem to have been band-passed? All this is rather unconventional. There is no reference provided to maybe see some alternative definition of PLV. Also, in the conventional definition of PLV shifting two signals by some time-lag will change the average of $\exp(\phi_1 - \phi_2)$ but not its absolute value. I can't quite decide if the problem here is with the description of the method, or the method itself. As written, I don't understand any of it.

Besides, the time-shift argument seems to suggest that you are measuring some sort of delay correlation, very similar to the TRF. So I really do not see what the added value is of PLV. It seems to me the paper would be better off without it.

<https://www.ncbi.nlm.nih.gov/pmc/articles/PMC6873296/pdf/HBM-8-194.pdf>

I have not read the Discussion, so can not comment on appropriateness.

Reviewer #2 (Remarks to the Author):

Title: "Speech-induced suppression during natural dialogues"

The authors present an EEG study where they assess the neural responses to spoken and heard speech. They find strong TRF responses to heard but not spoken speech, suggesting stimulus-induced-suppression. The experiment is interesting and the hypothesis reasonable. The narrative is very good, making the manuscript very enjoyable to read. However, there are substantial issues with the methodology and missing descriptions that make me doubt the results and interpretation. Key pieces of methodology that have not been reported. A risky double-filtering operations in preprocessing (as well as the details of those filters) was carried out without justification. Very large EEG prediction correlations are measured (about 5 or 10 times what is typically seen in the literature), without a proper explanation on how those correlations were calculated. Furthermore, one key issue is the denoising. "Good" auditory responses likely correlate with the removed "motor" activity. Could that explain why the TRF to produced speech is very small? These main issues prevent publication of this work in its current form. The missing methodological details might clarify the paper and convince me of the validity of the procedure. However, as it stands, the paper cannot be accepted for publication in my view. Please find my comments below.

Major comments

1. Results in Fig. 2.3B and D. Could it be that ICA is simply removing too much from the EEG data? This is a very important issue that should be discussed, as it mines the whole point of the paper.
2. Line 93: It's unclear how such high prediction correlation values are obtained. Even the paper cited by the authors shows maximum EEG prediction correlations of around 0.05 (not 0.5!). The specific section of the methods does not clarify this aspect. That must be clarified.
3. Line 310: So, does that mean that you re-filter data that was filtered already. That is problematic. As you don't report the details of the second round of filters, it is impossible for me to estimate the impulse/step responses of the series of filters, which is worrying. Also, are we talking about zero-phase shift filters? Please clarify.
4. Was the data re-referenced to the mastoids (I assume so, but it's not indicated in the text)? The TRFs don't show much polarisation (how does the GFP look like?). Is that all that response coming from the mastoids for some reasons? How do the results look like with a different referencing.

Other comments

5. Abstract. It is unclear what the authors mean with "better performances" (Stronger effects of SIS?)
6. Line 79: It's unclear if the authors intend to analyse perception only at first or production. I assume perception, since the text says that they are trying to replicate previous work initially (but the text does not indicate which exact previous work we should refer to among the many ones cited above)
7. Line 60: True, but is imprecise. The parameters are called TRF because they estimate the TRF. But there are many ways to estimate a TRF. To be fair, even a cross-correlation is an estimate of the TRF (even if less accurate). This is me being picky here. This one is not important.
8. Fig. 2.2C. The ylabel is incorrect. I should be "Frequency (Hz)"
9. Fig. 2.2B-Right. That is visually confusing and not convincing, honestly. But I won't go into detail until it's clear why those prediction correlation values are so high
10. Line 111 and 312. Broad ERP band? Do you mean "EEG"? Or simply broader-band?
11. The caption of figure 2.2 could be clearer. It should stand on its own. Instead, it's unclear what

"frequency band" refers to (in this case, EEG frequency band) and what the correlaton is calculated on.

12. How were the electrodes selected for the lateralisation analysis

13. Line 314: It's fine that you explain that you used MNE for the filtering. However, you should indicate the version of MNE, as well as all the details of the filters (not just what function you used).

14. Line 309: What do you mean with "supervised by an expert"? Was that one of the authors? Could you not simply say "manually detected"? Also, are there some parameters that one should specify with those functions (as you did on line 303)? Keep in mind that this must be replicable.

15. Line 130: "An absence of response" s a tricky thing to say, statistically speaking. Maybe you mean that no TRF component was significantly above chance?

16. Figure 2.3 should be improved from a graphical perspective (e.g., hard to read, labels not aligned).

17. How is the "silence" TRF calculated? i.e., what is the stimulus vector in the encoding model?

18. The TRFs look delayed compared to previous studies (using encoding models). The N1 usually emerges around 80ms, and the P2 around 140-160ms (in adults). However, the only discussion about that seems to be on lines 115 and 204, where the TRF is compared with ERPS instead.

19. Lines 186 and 205: That statement is incorrect. A few studies focused on the 1-15Hz band, but there is a lot of variability in the field in this regard. Also, please note that on of the two studies included as a reference to support that statement actually looked into the various sub-bands.

Furthermore, saying that theta-band shows "greater representation" of acoustic features s debatable. You could say that the prediction correlations are larger than in other bands. But that could mean many other more methodological things. Also, line 209: "main frequency band" and "this could be due to the cortical entrainment effect". All these statements are vague and, frankly, it is unclear what the authors are trying to say.

20. Line 221. This is EEG. It is fine if the authors want to speculate on that, as long as it's clear that it is only a speculative reflection. However, I recommend not to go down that rabbit hole. This is only EEG!

21. Line 176. This paragraph is absolutely misleading. Those numbers are from different metrics. As far as I understand, some of that work uses a re-scaled prediction correlation (based on an ideal maximum estimated based on the trial-by-trial noise), while others do not. That discussion has to be corrected and the methods regarding this specific paper should also be much clearer about the procedure here.

22. Line 192: That also depends on how the envelope and spectrogram are calculated.

23. What happens for pre-stim lags? It is unclear why that is not included. This is particularly relevant as we are talking about speech production. Pre-stim responses would not only be related to muscle movement and motion! That would also be a further validation on the author's assumption that their denoising is actually removing the motion/motor components.

24. I might have missed it. Is there a data sharing statement?

25. Some details are missing in the caption for Figure 2.5.

Reviewer #1 (Remarks to the Author):

Comment: This is a well conceived study on the difference in neural responses to speech of others vs our own speech. The introduction provides a good background discussion and motivation. The main finding appears consistent with prior literature showing reduced EEG responses to our own speech. The results, if confirmed, would generalize artificial laboratory tasks to natural dialog. Unfortunately the study has a small sample size (N=9 pairs of speakers/listeners) and the main result – the contrast in neural responses while both individuals are speaking – is uncertain, because statistics are not adequately performed. Generally, the manuscript is lacking in presentation of the statistics and analysis methods. Perhaps this can be fixed with a thorough rewrite of the Results section, but the small sample size will remain a hurdle.

Response: We greatly appreciate the reviewer's constructive feedback and time. In the following we provide a point-by-point response to each of their comments, highlighting in blue the major changes in the manuscript. We strongly believe that the manuscript has significantly improved its quality after taking into account the reviewer's comments.

Comment: In the following are comments to text in the order it appears in the manuscript.

"We were able to reproduce previous results and even slightly improve the performance of previous models using naturalistic stimuli (Fig. 2.2). We obtained a maximum correlation value of 0.57 for the envelope and 0.67 for the spectrogram, where the best previous correlation values were approximately 0.5 for the envelope [23]."

The values reported in Fig. 2.2 are indeed high. However, the presentation of the results is lacking. Neither caption nor text describe the results adequately. This is true for most of the Results section. Fortunately all this has an easy fix. Just say what the results are: No where in the figures can one see values 0.57 or 0.67. Instead, it appears from panel B that values max Fout below 0.41 and 0.28. Besides, reporting maxima is not particularly insightful as maximum values are inherently noisy. Panel A shows distributions, but it is not said over what these distributions are taken (electrodes? Subjects?). If it is over subjects, was the mean or maximum taken across electrodes for each subject? It is also unclear what the different EEG bands mean. Are you modeling the power of those signals, or the filtered signals itself? The methods only say that you filtered in those bands.

Response: We thank the reviewer for their careful lecture of our manuscript. The encoding models were used to fit and predict the filtered EEG signals (using the standard band limits and names). Following the reviewer's suggestion, we split our previous Figure 2.2 into Figure 2.2 and 2.3 in the current version of our manuscript, for easier and more complete description. The new Figure 2.2-A shows the distribution of correlation values obtained between the real and the predicted EEG signal for each electrode. These correlation values were first averaged across the 5 folds within each participant and then averaged across participants, obtaining in the end one value for each electrode. The top panel A shows the spatial distribution of those correlation

values to better illustrate the regions where higher correlations are achieved. On the other hand, the distribution shown in the lower part of panel A consists of those same values but presented as a violin plot (128 values per violin, each corresponding to a different electrode) for an easier comparison across frequency bands. This plot aims to provide an easier comparison between the correlation obtained when predicting different EEG bands.

The maximum values reported in the text corresponded to the highest 5-fold-average correlation values obtained for an individual participant and electrode, thus cannot be seen directly in the plots, where the average is also taken from all participants. This maximum value was reported for the sake of comparison with a previous highly related work [Desai2021preprint], where the authors present the correlation maximum value of ~ 0.5 for the band 1 - 15 Hz (see Fig. 2 from [Desai2021preprint]). The maximum value is usually reported because it is not expected that the speech signal impacts all the electrodes in the same way, and thus averaging correlation values across 128 electrodes could largely shadow the results. Following the reviewer's comments, we have updated the values by reporting the subject averages and maximum values, in the final version (lines 100 - 103).

From reading the rest of the paper, I am guessing you are not attempting to predict power in those bands, but instead the filtered signal itself in the band. That is a fine idea, but there is no need to estimate new TRF for each of these filtered signals. TRFs can capture all frequencies in an input→output mapping. There is no need to filter the output in different frequency bands. Typically the way you do this is to find a single mapping to the broadband signal. To measure how well each frequency band was predicted you can just measure the coherence spectrum between predicted output and real output, i.e. eeg. This is quite conventional linear systems. The only caveat is that regularization has a nonlinear effect on the estimate. So different bands may be variably affected, and so the TRF (vs the decomposed TRFs) may yield somewhat different results. But it does reduce the number of free parameters (by a factor of 5, to be exact, as you are using 5 different versions of the same signal in your current approach, broadband + 4 pass-band filtered version). Given the small sample size of this study, this may be helpful.

Response: As the reviewer correctly pointed out, the predictions are indeed realised on different filtered EEG signals, which is standard practice for analysing EEG. Aiming to better show the Speech Induced Suppression (SIS) effect during natural dialogues, we focus only on the frequency band with higher response, filtering out as much noise as possible. Thus, analysing the EEG data only in the Theta band for the SIS effect gives us more statistical power.

As it can be seen in Fig. 2.2A the correlation obtained using the broad-band (1 - 40 Hz) is significantly lower, with an average of 0.17, compared with the one obtained using the same EEG signal filtered in the Theta band (4 - 8 Hz), 0.37. Moreover, the Temporal Response Function (TRF) depicted in Figure R1 (see below), only presents a small peak (~ 0.0025) close to 0ms and lacks the expected waveform. Similar results were obtained for the [1, 15] Hz band used in other studies [DiLiberto2015, Desai2021]. This is not shown in the manuscript, as the focus is on the SIS effect.

broad-band (1-40 Hz)

Figure R1. mTRF for all electrodes for the spectrogram model fitted to the broad-band (left) and a 1-15 Hz band (right). Both broad-bands present a lower amplitude (~ 0.003 , ~ 0.009) peak close to zero, compared to the theta band (~ 0.014).

Concerning the sample size, this study has 9 dialogue sessions, with 17 to 30 trials per session (average of 24.3 trials). Each trial has a duration between 1 to 5 minutes, with an average of 82.6 seconds (SD=61s). This results in a total of 18 different participants. Each participant had their own unique stimuli (i.e. distinct self/interlocutor's speech pairs), and their own EEG recording. Thus, we have 18 independent samples for the most part of our statistical analysis; in particular, the comparisons between conditions were performed across participants, over the correlation values estimated for each participant (Fig. 2.4, 2.5 in the new version). Compared with the literature, our number of participants is actually higher than the ones reported by other highly related works in this area, for instance: Desai et al. (2021): 17 participants, Etard et al. (2019): 12 participants, Di Liberto et al. (2015): 10 participants.

Moreover, the estimation and validation of the model was performed at the participant level, obtaining significant results for all participants, assessed by a permutation test. Figure D.1 B in the Supplementary materials shows the results of the permutation tests for each band and

input feature. Panel C presents the distribution of the number of participants that passed the permutation test for each electrode (see Figure R5 and the new figure caption below for a more detailed explanation of this figure).

Panel 2.2 C top shows a scalp plot that is not explained. The time courses are not clear. There are multiple curves, are these for multiple electrodes or a single electrode and multiple subjects? Or perhaps each curve is for one of the different spectrogram features? The caption of figure 2.3 seems to suggest that. It should really be made clear already in figure 2.2. They look all very similar, maybe they can all use the same TRF and thus have the same number of parameters with the envelope, so the correlation coefficients are comparable. As it is, the higher values with the spectrogram (Figure 2.2B) could simply result from the larger number of free parameters. How exactly were the different bands in the spectrogram combined?

Response: Figure 2.3-A (in the new version of the manuscript) shows the mTRF for the spectrogram feature. It was obtained by fitting a model to each participant and then averaging the resulting mTRFs. Given that the spectrogram feature has multiple dimensions (16 mel-frequency bands), the data from each band was concatenated and one model for each channel was fitted. As the reviewer correctly pointed out, the spectrogram model had 16 times the number of features, and therefore 16 times the number of free parameters, compared with the model fitted using the envelope signal. Therefore, the obtained mTRF for each electrode is a $N_{delays} \times N_{bands}$ 2D matrix, and then they are aggregated across electrodes to build a $N_{delays} \times N_{bands} \times N_{electrodes}$ 3D matrix.

Figure 2.3 shows different representations of this matrix. Panel A presents the results averaged across frequency bands where each electrode is presented as a coloured curve and the electrode positions are referenced in the colour scalp plot ($N_{delays} \times N_{electrodes}$ matrix). Panel B presents the results averaged across electrodes depicted as a heatmap ($N_{delays} \times N_{bands}$ matrix).

As the reviewer correctly states, it is possible that the spectrogram model outperforms the envelope model because of the greater number of free parameters involved. Nevertheless, the regularisation parameter was computed for each feature separately. It took greater values (1325 vs 976) for the spectrogram compared to the envelope, which could contribute to mitigating the effects of a greater number of coefficients. Furthermore, the comparison made in Figure 2.2 B is made within each feature (spectrogram and envelope). The difference between the correlations obtained for each feature are out of the main scope of the present manuscript. They are discussed here to make a comparison with the bibliography.

Figure 2.2 B aims to show how different features (spectrogram and envelope) present different lateralization between frontal left and right electrodes, represented in the blue and orange box plots respectively. These 12 electrodes were selected because they present the highest correlation in each region. Further details for this experiment are explained in the next response.

Following the reviewer's suggestion, we added more detailed captions in Figures 2.2 and 2.3 in the new version of our manuscript, aiming for a more clear interpretation of the graph.

"In the Theta band, the representation of the spectrogram is stronger in the frontal lateral regions of both hemispheres, more lateralized to the left, where language-related areas such as Broca are found (Fig. 2.2 B; Wilcoxon test: $z=12$, $n=12$, p -value = 0.034)."

How was this set of electrodes selected? The p-value here is borderline, if more than one set of electrodes was tested, this clearly is no longer meaningful. Also, the word "representation" is something you can propose in the discussion, but it's not the data. What you probably mean to say is that the spectrogram predicts those electrodes best, or that the correlation is highest for those electrodes. It does not mean that this part of the brain "represents" the spectrogram. Indeed, low level features like that are likely best "represented" in the auditory cortex at a much faster timescale. Why the theta (power?) is more robustly driven by the spectrogram features in those frontal electrodes is worth a discussion, in the Discussion section.

Response: The electrodes were selected by visual inspection, based on the regions with higher correlation values (symmetric on both hemispheres). These regions partially matched the regions reported in [Etard2019, DiLiberto2015] and intracranial EEG studies [Tang2017, Hamilton2018]. Nevertheless, the result does not critically depend on the exact electrode selection. Aiming to clarify this valid reviewer's concern, we performed new experiments to better demonstrate the mentioned effect. We repeated the analysis, but selected different combinations of channels based on a different number of higher correlation values (and all frontal lateral electrodes). The new results are depicted in the following figures.

Figure R2. Spectrogram response lateralization on the theta band. The top panel shows the average correlation values distribution. The bottom panels show correlation distribution for left and right electrodes indicated in the topographic figure, for the spectrogram model for 8, 10, 12, and 22 selected electrodes. The electrodes were chosen in each case, as the ones presenting higher correlation values in the frontal region for each hemisphere. A signed-rank Wilcoxon test was performed to compare the values obtained in each hemisphere. The correlation values for the spectrogram show a significant lateralization effect towards the left hemisphere in all cases. Significance: ** p-value < 0.01, *** p-value < 0.001.

Figure R3. Envelope response lateralization on the theta band. The top panel shows the average correlation values distribution. The lower panels show the correlation distribution for left and right electrodes indicated in the topographic figure, for the envelope model for 8, 10, 12, and 22 selected electrodes. The electrodes were chosen in each case, as the ones presenting higher correlation values in the frontal region for each hemisphere. A signed-rank Wilcoxon test was performed to compare the values obtained in each hemisphere. The correlation values for the envelope show no significant lateralization effect when considering the higher correlation values, but it does when considering all electrodes in the frontal lateral region. Significance: n.s. p-value > 0.05, * p-value < 0.05.

Following the reviewer's suggestion, we replaced the term 'representation' in the Results section, and discussed it in the Discussion section, line 210.

"It is possible to observe a gradual lateralization effect from frontal electrodes in higher frequency bands to more lateral in the Theta band (Fig. 2.2 A)." I am sorry, but I can't see that in figure 2.2A. And even if there was a subjective impression on those figures, colormaps can be incredibly deceptive. I would like to see some sort of statistical analysis to support this observation. Otherwise it's just tea leafing.

Response: Following this reviewer's suggestions, we replaced our previous claim by *"It is possible to observe a lateralization effect in all frequency bands (Supplementary Fig. C.3)."* in lines 112-113 of the new version of our manuscript. Moreover to better back up our findings, we performed a new analysis where we observed the same lateralization using the spectrogram features in the other EEG frequency bands. In Figure. R4 we can observe that the lateralization effect is significant across all frequency bands used for the analysis, with p-value lower than 0.0009 in all cases.

Figure R4. Spectrogram response lateralization across frequency bands. The top panels show the selected number and positions of frontal electrodes with higher correlation values on each hemisphere. The bottom panels show the difference in the correlation values between hemispheres. Significance: *** p-value < 0.001.

“The number of subjects for which the obtained correlations were significantly different from the random distribution in all folds was ...” In the methods this test was performed separately per electrode. But here there is only talk of subjects. Do you mean where at least one electrode generated significant correlation? Or how else are these numbers established? Incidentally, there are only 9 pairs of subjects. The numbers here suggest that this was done for 18 subjects. Is that right? Also, it is not clear if this analysis is for the spectrogram features or the envelope.

Response: The statistical analysis made in Section 2.1 was done to assess the performance of the model on each participant. This also allowed us to endorse the significance of the results about the representation of acoustic features in the EEG signal. To that end, a permutation test, described in Section 4.7.1, was performed for each electrode of each participant separately. This test was repeated over the 5 folds of the cross-validation.

An electrode was considered to have a significant effect if its correlations passed the permutation test in all folds. After this analysis, each electrode is marked as significant or non-significant for each participant. To summarise this process for all the participants, we can determine in how many participants each electrode was considered significant or not, where the maximum possible value is 18, as the number of participants. The scalp plots from Supplementary Figure D.1-C show the number of participants where the electrodes are considered significant for the spectrogram model.

Figure R5. Spatial distribution of model significance across participants for every frequency band of the spectrogram feature, corresponding to the first row of panel B. A permutation test was applied to each electrode, fold, and participant (see Section 4.7.1). An electrode was considered as having a significant effect if its correlations passed the test in all the folds. The scalp distributions show the number of

participants with significant results for each electrode. The maximum possible value was 18, as the number of participants.

To summarise the results in the text, we averaged these values (i.e. the number of significant participants per electrode) across electrodes, and presented them as the mean \pm s.e.m in lines 117-121.

“non-altered evaluation set” I don't know what you mean by “altered”. You mean “altered”? What was altered in the rest of the data, but not on the “evaluation set”? And in the “random permutation” what exactly was permuted? Details here matter. As with all random stats, if correlation structure in the data is perturbed (except for the one specifically to be tested) the Null distribution can be quite incorrect.

Response: We thank the reviewer for bringing up this important topic. In the previous version, the “non-altered evaluation set” referred to the “original data”. Please, find a more detailed explanation of the permutation procedure below.

The input matrix for the model consisted of ($N_{samples}$) rows and ($N_{times} \times N_{features}$) columns. Each sample corresponded to an interval where the participant had been uninterruptedly listening to their partner speak for at least 0.6 seconds (condition E in our manuscript). Using overlapped sliding windows with 1 time-point step, all the valid intervals within each session were extracted (for each condition separately). Around 50,000 samples per participant for the E condition were obtained (Fig. R6.A,B).

As the EEG and audio sampling rates were both 128Hz, each interval (or sample) of 0.6 seconds contained 77 time-points ($n_{times}=77$). In the case of the spectrogram, as there are 16 frequency bands, the $n_{times} \times features$ correspond to a $77 \times 16 = 1232$ vector.

The permutations test was implemented by making 3000 random permutations of the input matrix. This analysis was performed for each participant, electrode and fold. The permutations consisted only in rearranging the samples, i.e. assigning the EEG interval to a different audio interval. Thus, these random permutations conserved the correlation structure between subsequent time-points (Fig. R6.C). A similar procedure was also implemented in [Desai2021] but with a much lower number of permutations (100).

The evaluation set kept its samples and time ordering, as in the original data.

Following the reviewer's comment, we explain in more detail this procedure in the 4.6 and 4.7.1 section of our manuscript.

Figure R6. Schema of the definition of samples, valid samples, and permuted samples.

“Moreover, within the spectrogram, the more important features are the audio mel-bands ranging from 583 to 1907 Hz, which correspond to the ones where human speech carries more information.” How was this established? If Figure 2.2C shows TRF to different frequency bands, they all look very similar. You would need some sort of statistical test to say one is bigger than another, perhaps do stats across subjects.

Response: We thank the reviewer for this comment. To statistically support the claim, we performed a threshold-free cluster-enhancement (TFCE) [Mensen2013] test on the mTRFs across participants and identified the significant cluster of the averaged response shown in Fig. 2.3B (2.2C in the previous version of the manuscript). We used the 1-sample permutations test implementation of MNE (https://mne.tools/stable/generated/mne.stats.permutation_cluster_test.html).

The new Figure 2.3-C shows the resulting p-values, where a significant cluster is present between 583 to 2281 Hz mel-bands, supporting our previous hypothesis.

We now include a new section in our manuscript, explaining the used methodology in page 22-23 (Section 4.7.3).

“The mTRFs present an absence of response in any frequency band in both situations, and are similar to the response to background noise when both participants are silent (Fig. 2.3 E). This was confirmed by a pairwise comparison between the correlation coefficients from (S) and (S|B) conditions with Silence.”

The lack of evidence for an effect is not the same as evidence for a lack of an effect. But that is not even what is interesting in this experiment. Suppression does not need to be so strong as to

not have a response at all to self speech. Additionally, poor correlation with simultaneous speaking may be due to noise in the EEG during speaking, which is substantial – something you acknowledge later in the result section.

Rather, what you want to show is that the response is weaker, i.e. a contrast between panels C and D. Namely, both are speaking, but the EEG response to the other speaker is stronger than the EEG response to self speech. This analysis you do in fact perform a little later. It is the central finding of the paper, as it controls for speaking noise in EEG, and shows an effect, rather than arguing for a lack of an effect, which is always difficult to establish. And this begins the “Fig. 2.4; S vs Silence: uncorrected p-value > 0.12 for all channels, mean uncorrected p-values = 0.52; (S|B) vs Silence: uncorrected p-value > 5.3×10^{-4} for all channels, mean uncorrected p-values = 0.22” If you really want to show evidence in support of the Null hypothesis, you should report Bayes factors.

Response: As the reviewer correctly stated, the comparison between panels C and D in Figure 2.4 are central to our manuscript, as those are the ones showing a much stronger response to the other speaker audio compared to the self-produced speech. This is supported by other results such as the comparison between panels D and E, Listening to the self-produced speech while both are speaking (S | B) and Silence, where no significant differences between them were found even though this could not be considered proof for lack of response, it shows a response more similar to the silent stimulus.

We want to remark that we did not aim to, or could, prove a complete lack of response, as we acknowledge that “*the lack of evidence for an effect is not the same as evidence for a lack of an effect*”. However, we believe that it is important to highlight that we showed not only a significant reduction of the response to the own speech in comparison with the response to the other’s speech, as in previous studies, but also that the response to the own participant speech is indistinguishable from Silence condition even with a stronger analysis approach (than ERPs). Silence condition is presented as a negative control condition.

In the discussion, we present two alternative hypotheses, the SIS effect could be driven by a corollary discharge mechanism or by a selective attention mechanism. Although we acknowledge that our results are not conclusive, we argue that the full attenuation is more consistent with a corollary discharge mechanism as proposed by the bibliography [Scheerer2013,Wang2014]. This mechanism was largely explored in other motor-sensory phenomena such as eye movements or tickles in which a complete attenuation is observed [Blakemore1998,Thiele2002]. Although several works also proposed a corollary discharge mechanism but observed on a partial attenuation [Curio2000, Houde2002], a partial attenuation could be more consistent with a selective attention hypothesis, in agreement with auditory or visual selective attention experiments [Power2012,Osullivan2015].

Regarding the noise, as discussed also by [Perez2022] the noise in this case would possibly increase correlation instead of disrupting it since, if present, it is associated with one’s own speech.

Following this reviewer's comments, we changed the term 'no response', for 'no evidence for response' in the results and method sections. Also, to better support our findings, and following a later reviewer's recommendation, we added Cohen's d-prime analysis in the current version of our manuscript, and FDR analysis in the supplementary section.

"(Fig. 2.4; uncorrected p-values $< 3.9 \times 10^{-4}$ for 84 channels; mean uncorrected p-values $< 4.4 \times 10^{-4}$)" It is never clear in this paper how statistical comparisons have been done. Are correlation values computed per subject and then the stats performed across subjects? The degrees of freedom or the sample size needs to be reported for each statistical evaluation in the results section. It's hard to imagine small p-values with N=18 subjects, and Bonferroni correction across 128 electrodes (as stated in the methods).

Response: The comparisons between the predicted and original signals were made for each electrode separately. Correlation values were estimated for each fold, electrode, participant, and condition separately. They were first averaged within folds yielding 18 values (one per participant) per condition and electrode. These values were paired between conditions.

We perform a Wilcoxon signed-rank test between two conditions for each electrode (using `scipy.stats.wilcoxon` function from `scipy 1.7.1`), resulting in 128 tests per pair of conditions. We corrected the threshold of significance using the Bonferroni procedure (equal to $0.05 / 128$) and we also added Cohen's d-primes to express the effect sizes. The results were presented as scalp distributions by illustrative means.

Regarding the low p-value, by observing Figure 2.4 one can appreciate that the correlation values in (E | B) for the Theta band are on average around 0.2 for all electrodes, whereas for (S | B) and Silence, the average is around 0.05. Given these considerable differences, it is not unlikely that the Wilcoxon test would yield such values, even with N=18. We now specified which function we used in the text. For a more practical example, we perform a simple experiment with a sample of N=18 values in which the p-value is as low as:

```
from scipy.stats import wilcoxon
import numpy as np
n = 18
a = np.random.rand(n)
b = np.random.rand(n) + 2
z,p = wilcoxon(a,b)
print("p-value = %.7f"%p) # p-value = 0.0000076
```

Along the manuscript, four different statistical tests were used. In the present version of our manuscript we expanded the explanation in sections 4.6 - 4.7 following the reviewer's comments. The statistical tests performed are the following:

1. The predictive capacity of each model (for each fold, electrode, and participant) was assessed using the permutation test, as explained in section 4.7.1. This test was

performed using both the spectrogram and the envelope as input features and using the different filter EEG frequency bands as target.

2. When comparing the lateralization effect, depicted in Figure 2.2B, we used a Wilcoxon signed-rank test (scipy.stats module v1.7.1) to compare the correlation values obtained for the left and right electrodes across participants (see section 4.7.2).
3. To identify the significant time-points and frequency bands for the fitting and prediction of the EEG signal across participants, we used a 1-sample permutations test (Threshold-Free Cluster Enhancement, TFCE) [Mensen2013] (see Fig. 2.3 and section 4.7.3).
4. For comparing the different listening conditions depicted in Figure 2.5, we used a Wilcoxon signed-rank test between correlation values for each condition across participants. We repeated this comparison for each electrode and corrected the threshold for multiple comparisons using Bonferroni's correction (see section 4.7.4.).

Also, comparisons rarely tell the reader what measures are being used? I am guessing mostly correlation with theta band EEG predicted from spectrogram? Or from an envelope?

Response: As the reviewer correctly points out, values presented in Figures 2.3 and 2.4 are the correlations measured using the spectrogram as an input feature and the EEG Theta band as the target, as this model is the one that shows better results. The envelope signal was used with the mTRF approach as a complementary analysis to compare with the literature, as it is commonly used in several related works [DiLiberto2015, Etard2019, Desai2021], and helped us validate that our model was properly working. The envelope is finally also used in the Phase-Locking value (PLV) analysis (Figure 2.6).

Following the reviewer's comments now we stated this more clearly in the Methods section 4.7.4, and in the caption of Figure 2.5.

Also, it is never explained if TRFs are computed per subject, or together for all subjects.

Response: The TRFs are computed for each electrode and participant separately, obtaining 128 TRF time series, and thus 128 correlation values, per participant. In the new version of our manuscript we further clarified this important point along with the statistical comparisons in the results section (lines 89-90) and Methods section 4.6.

Now to the actual result shown in figure 2.4 "This was statistically confirmed by pairwise comparison between the correlation values from (E|B) and (S|B) when both participants were speaking simultaneously, ... (Fig. 2.4; uncorrected p-values $< 3.9 \times 10^{-4}$ for 84 channels)."

To me, this is the central result of the paper, so it deserves some attention. The relevant scalp plot in figure 2.4 looks rather noisy in the sense that neighbouring electrodes have quite different p-values. Real brain activity is expected to have smoother spatial distribution due to the blurring function of CSF and scalp. P-values also tend to be noisy, certainly on N=18. My first suggestion is to do a proper multiple comparison correction, like FDR. That 84 of 128 channels are below some arbitrary hand-picked p-value is not appropriate statistical comparison or reporting. The second recommendation is to show the effect size, rather than p-values, for instance, Cohen's

d-prime. That will be a lot more informative, and likely smoother in space. My guess is that with N=18 and 128 electrodes, you are not adequately powered to detect statistically significant differences.

Response: We thank the reviewer for taking special attention to the central finding of our manuscript. We decided to choose Bonferroni's correction as it is the most rigorous correction. Although we acknowledge that Bonferroni assumes independence between electrodes, which is more strict than what we have, having significant results with such correction seems very robust.

The FDR test is a less conservative approach, compared with Bonferroni's correction, that allows false-positives and it does not take into account the spatial dependence between electrodes. Still, following the reviewer's suggestion, we performed a FDR and Cohen's d' analysis. The results are depicted in Figure R7 below. As expected, the scalp distributions of the p-values present a similar pattern of results that the one present in Figure 2.5-A using Bonferroni's correction. As Bonferroni's correction is more rigorous analysis we strongly believe that it better represents our results. We decide to include the Cohen's d-primes for the effect sizes.

The 3.9×10^{-4} value result for correcting the threshold with Bonferroni ($0.05 / 128$). Now we changed the reference to p-values as 'uncorrected p-values'. We hope the procedure is clearer now.

Figure R7. Comparison between listening conditions. **Top left:** Cohen's d-prime for the distribution of average correlation values of each electrode from different conditions in the Theta band (N=18). **Top Right:** FDR corrected p-values and **Bottom:** Uncorrected p-values from a Wilcoxon signed-rank test, channel-by-channel, between the average correlation values of each electrode from different conditions in

the Theta band (N=18; see Fig. 2.3). The p-values were corrected using FDR and non-significant electrodes. The conditions are abbreviated as follows: Listening to external speech (E), Listening to self-produced speech (S), Listening to the external speech while both are speaking (E|B), Listening to the self-produced speech while both are speaking (S|B).

“This method averages single tone vectors in the complex plane, where the phase of those vectors corresponds to the phase difference between signals at each time (t). ... This analysis was repeated time-shifting the envelope signal ... to find the time-lag when the synchronisation between signals is maximum.”

PLV -- as introduced in Lachaux 1999 (reference below) – averages phase angle across trials. It is not clear what was done here. What are “single tone vectors”? And over what are averages computed, i.e. what exactly is t in equation (2)? Given the argument about time-shift, it seems the average was taken across time samples within utterances? Multiplying the angle difference with t makes little sense to me. Also, how are angles computed? Usually Hilbert transforms of some band-passed signal. But here the stimulus (envelope?) does not seem to have been band-passed? All this is rather unconventional. There is no reference provided to maybe see some alternative definition of PLV. Also, in the conventional definition of PLV shifting two signals by some time-lag will change the average of $\exp(\phi_1 - \phi_2)$ but not its absolute value. I can't quite decide if the problem here is with the description of the method, or the method itself. As written, I don't understand any of it.

Besides, the time-shift argument seems to suggest that you are measuring some sort of delay correlation, very similar to the TRF. So I really do not see what the added value is of PLV. It seems to me the paper would be better off without it.

<https://www.ncbi.nlm.nih.gov/pmc/articles/PMC6873296/pdf/HBM-8-194.pdf>

Response: We agree with the reviewer that the PLV has similar implications than the TRF. But, being a different implementation, we had added it as a model free confirmation of the results obtained by the TRF. In addition, it also brings information about the latency of the synchronisation to the heard audio envelope. Please, find a more detailed explanation of the PLV implementation and its methodological aspects below.

Single tone vectors referred to the “unit vectors”, meaning phase angles, represented as unit vectors in the complex space. We replaced the term “single tone vector” to “unit vectors” in the new version of the manuscript to follow a terminology consistent with the bibliography. The phase of those unit vectors corresponds to the phase difference between the envelope and a given electrode signal.

“...where the phase of those vectors corresponds to the phase difference between signals at each time (t).” As the reviewer points out, “t” stands for time, but it is not multiplying in the equation (2), but only a sub index indicating the phase difference between those angles at a given time “t”. We changed the equation (2) for a more clear visualisation and interpretation. We averaged these phase difference vectors across all the time samples for every condition as one unique trial. For example, to calculate the phase synchronisation from time lag 0 in one

condition, all the samples of the EEG at that condition were taken, and used in the PLV equation alongside with the samples from the envelope corresponding to the same time. To calculate the phase synchronisation at time lag 200 ms, the EEG samples from that condition were used in the PLV equation, but this time with the envelope signal delayed 200 ms.

This PLV implementation consists of only one trial using the samples corresponding to each condition ((S), (E | B), ...) on which we compare the phase of the envelope signal and the EEG signal of each electrode. As the reviewer points out, this method is traditionally implemented for epoched data, obtaining the synchronisation of multiple signals (often from different sensors) over time along the trial. But also, it is often used as a connectivity measure, indicating the synchronisation between pairs of signals. In that case, the PLV is computed for pairs of signals across whole trials, and the results are then averaged over trials to yield one measure of connectivity for each pair of signals. The analysis implemented in our manuscript is similar to this one, but considering the whole signal as one trial. We used trials of 0.6 seconds of envelope and EEG signals to compute the synchronisation over each trial, and then average the results over trials that would have been possible. However, having a PLV value determined by the whole signal as a trial rather than an average of averages seemed more rigorous.

As the reviewer suggests, the phase angles were computed using the Hilbert transform of the Theta filtered signal of the EEG electrodes, and the Hilbert transform of the envelope signal without any filter. It is true that this method traditionally filters both signals in the same frequency bands, and we corrected that in the new version. We implemented a 3rd order butterworth filter with *scipy* *butter* and *filtfilt* functions between 4 and 8 Hz on the envelope signal (to prevent phase distortions in the envelope signal), and then extracted the phase from the analytic signal with the Hilber transform. The results are consistent with our previous results, but with higher values. We thank the reviewer for this suggestion, as it is an improvement that brings a more correct implementation of our method.

With regards to the affirmation that a phase difference would not change the absolute value of the average of imaginary exponentials, we are sorry but we are not sure we understand. If two signals were pure sines with one frequency, then yes, the phase difference would be constant, and the absolute value of the average of the imaginary exponentials (PLV) would not depend on the phase difference. That would happen because all the imaginary unit vectors would be aligned, and their average would have an amplitude of 1. However, with more complex signals, time lags introduce changing correspondence between the two phase vectors, and thus the phase differences over time, making the resulting imaginary unit vectors spread through the complex plane, and the resulting average of those vectors would have different absolute values.

The subscripts are very small in the equation, now we slightly changed the notation to make it clearer:

$$PLV(\tau) = \frac{1}{n} \left| \sum_{t=1}^n \exp(i(\theta_{env}(t) - \theta_{EEG}(t - \tau))) \right|$$

I have not read the Discussion, so can not comment on appropriateness.

Response: We highly appreciate the reviewer's time for reading the rest of the manuscript.

References:

- [Blakemore1998] Blakemore SJ, Wolpert DM, Frith CD. Central cancellation of self-produced tickle sensation. *Nature Neuroscience*. 1998;1(7):635-40.532
- [Desai2021] Desai M, Holder J, Villarreal C, Clark N, Hoang B, Hamilton LS. Generalizable EEG encoding models with naturalistic audiovisual stimuli. *Journal of Neuroscience*. 2021;41(43):8946-62.
- [DiLiberto2015] Di Liberto GM, O'Sullivan JA, Lalor EC. Low-frequency cortical entrainment to speech reflects phoneme-level processing. *Current Biology*. 2015;25(19):2457-65.
- [Etard2019] Etard O, Reichenbach T. Neural speech tracking in the theta and in the delta frequency band differentially encode clarity and comprehension of speech in noise. *Journal of Neuroscience*. 2019;39(29):5750-9.
- [Hamilton2018] Hamilton, L. S., Edwards, E., & Chang, E. F. (2018). A spatial map of onset and sustained responses to speech in the human superior temporal gyrus. *Current Biology*, 28(12), 1860-1871.
- [Perez2022] Pérez, A., Davis, M. H., Ince, R. A., Zhang, H., Fu, Z., Lamarca, M., ... & Monahan, P. J. (2022). Timing of brain entrainment to the speech envelope during speaking, listening and self-listening. *Cognition*, 224, 105051.
- [Thiele2002] Thiele A, Henning P, Kubischik M, Hoffmann KP. Neural mechanisms of saccadic suppression. *Science*. 2002;295(5564):2460-2.
- [Curio2000] Curio G, Neuloh G, Numminen J, Jousmäki V, Hari R. Speaking modifies voice-evoked activity in the human auditory cortex. *Human brain mapping*. 2000;9(4):183-91.539
- [Houde2002] Houde JF, Nagarajan SS, Sekihara K, Merzenich MM. Modulation of the auditory cortex during speech: an MEG study. *Journal of cognitive neuroscience*. 2002;14(8):1125-38.541
- [Scheerer2013] Scheerer NE, Behich J, Liu H, Jones JA. ERP correlates of the magnitude of pitch errors detected in the human voice. *Neuroscience*. 2013;240:176-85.543
- [Wang2014] Wang J, Mathalon DH, Roach BJ, Reilly J, Keedy SK, Sweeney JA, et al. Action planning and predictive coding when speaking. *Neuroimage*. 2014;91:91-8.

New references:

- [Desai2021preprint] Maansi Desai, Jade Holder, Cassandra Villarreal, Nat Clark, Liberty S. Hamilton "Generalizable EEG encoding models with naturalistic audiovisual stimuli" bioRxiv 2021.01.15.426856; doi: <https://doi.org/10.1101/2021.01.15.426856>
<https://www.biorxiv.org/content/10.1101/2021.01.15.426856v1>
- [Mensen2013] Mensen, A., & Khatami, R. (2013). Advanced EEG analysis using threshold-free cluster-enhancement and non-parametric statistics. *Neuroimage*, 67, 111-118.
- [Tang2017] Tang, C., Hamilton, L. S., & Chang, E. F. (2017). Intonational speech prosody encoding in the human auditory cortex. *Science*, 357(6353), 797-801.

- [OSullivan2015] O'sullivan, J. A., Power, A. J., Mesgarani, N., Rajaram, S., Foxe, J. J., Shinn-Cunningham, B. G., ... & Lalor, E. C. (2015). Attentional selection in a cocktail party environment can be decoded from single-trial EEG. *Cerebral cortex*, 25(7), 1697-1706.
- [Power2012] Power, A. J., Foxe, J. J., Forde, E. J., Reilly, R. B., & Lalor, E. C. (2012). At what time is the cocktail party? A late locus of selective attention to natural speech. *European Journal of Neuroscience*, 35(9), 1497-1503.

Reviewer #2 (Remarks to the Author):

Title: "Speech-induced suppression during natural dialogues"

The authors present an EEG study where they assess the neural responses to spoken and heard speech. They find strong TRF responses to heard but not spoken speech, suggesting stimulus-induced-suppression. The experiment is interesting and the hypothesis reasonable. The narrative is very good, making the manuscript very enjoyable to read. However, there are substantial issues with the methodology and missing descriptions that make me doubt the results and interpretation. Key pieces of methodology that have not been reported. A risky double-filtering operation in preprocessing (as well as the details of those filters) was carried out without justification. Very large EEG prediction correlations are measured (about 5 or 10 times what is typically seen in the literature), without a proper explanation on how those correlations were calculated. Furthermore, one key issue is the denoising. "Good" auditory responses likely correlate with the removed "motor" activity. Could that explain why the TRF to produce speech is very small? These main issues prevent publication of this work in its current form. The missing methodological details might clarify the paper and convince me of the validity of the procedure. However, as it stands, the paper cannot be accepted for publication in my view. Please find my comments below.

Response: We greatly appreciate the reviewer's time and constructive evaluation. We strongly believe that the reviewer's comments have significantly improved the manuscript's quality. Please, find a point-by-point response to each of their comments below and the major changes in the manuscript highlighted in blue.

Major comments

1. Results in Fig. 2.3B and D. Could it be that ICA is simply removing too much from the EEG data? This is a very important issue that should be discussed, as it mines the whole point of the paper.

Response: We thank the reviewer for pointing out this important issue. We do not restrict movements during the task, aside from having an EEG cap, remaining seated, having a blanket between them (to discourage facial and hand gestures) and asking to minimise unnecessary movements. Thus, during preprocessing we aimed to remove as much unrelated noise as possible without affecting neural signals.

We use Independent Components Analysis (ICA) mainly to remove eye-movements (using EyeCatch [Bigdely-Shamlo2013] and ADJUST criteria [Mognon2011]) and discontinuities that correspond to bad electrodes or isolated electrodes with high noise for an interval of time (using ADJUST criteria [Mognon2011]). Both methods recommend that the researchers supervise the selected components, and sometimes we include a rejected component, for instance when it showed a peak in alpha frequencies. The same components were used for the whole EEG recording, independently of whether the participant was speaking or listening, thus we do not expect correlations with the spectrogram or the envelope for these components introduced by bias on component selection.

We also removed other muscular artifacts. We identified these components as having sharp spatial distributions located in the edgemoost temporal electrodes (over the ears), with high frequency spectra (typically flat or U-shaped spectra), and without a peak in alpha. These components are usually just referred to as ‘muscle components’ in the bibliography [Tran2004, Muthukumaraswamy2013, Jansen2020] and tutorials on artifact removal with ICA (for instance https://eeglab.org/tutorials/06_RejectArtifacts/RunICA.html). They more likely capture neck or jaw movements, but not lips or tongue. The lips and tongue movement certainly could have a spectra more concentrated in lower frequencies, but an occipital-frontal dipole [Porcaro2015]. Furthermore, over the different articulators, the jaw is probably the one with lower correlations with the speech spectrogram [Chartier2018]. Thus, we do not expect the ICA preprocess to remove much of the spectrogram responses and thus introduce a potential bias towards lower responses in the speaking condition. On the contrary, as we are not removing components specifically related to tongue or lips artifacts, in the worst case, we could expect some immediate increase in the response in that condition, as discussed by [Perez2022].

It is important to note that the artifact removal was performed in the continuous data, before the separation into conditions. Overall, we removed 22 out of 128 components per participant. 17.5 using EyeCatch and ADJUST, and 4.5 using the following criteria for muscle artifacts: “spatial distributions with sharp peaks located in the edgemoost temporal electrodes (over the ears), with high frequency spectra (typically an U-shaped spectra), and without a peak in alpha.”.

We updated the information of the artifact correction in Section 4.3, pages 18-19 .

2. Line 93: It's unclear how such high prediction correlation values are obtained. Even the paper cited by the authors shows maximum EEG prediction correlations of around 0.05 (not 0.5!). The specific section of the methods does not clarify this aspect. That must be clarified.

Response: We thank the reviewer for pointing this out. Our prediction values are obtained by computing the correlation between the real and the predicted EEG signal, for all the samples from the corresponding condition (E: listening to the other participant speak for at least 0.6 s; etc...). To do so we follow the following steps: First, an encoding model was fitted for each participant and each electrode, obtaining one mTRF and a predicted signal for each electrode and participant. Then, Pearson's correlation values between the original and predicted signals were computed, yielding one correlation value per electrode and participant. As this procedure was repeated in a 5-fold cross-validation loop, the 5 correlation values of every electrode were averaged to report one correlation for each electrode and participant. These fold-averaged correlation values reached a maximum of 0.67 for one electrode of one participant, and they were reported as a comparison to the results obtained in [Desai2021preprint], where correlation values were displayed in Figure 2. From that plot, the maximum correlation value for one electrode was approximately 0.5 (for the envelope feature). Moreover, we argue that it is important to report maximum correlation values (over the electrodes) because it was not expected that all the 128 electrodes have strong responses to the stimulus and thus, the effect

could be washed out by the average. We now included not only maximum values but also mean values (even though mean values have the disadvantage of averaging over not-necessarily relevant electrodes).

Di Liberto et al. (2015) presented correlation values obtained for a broad-band (1-15 Hz) of around 0.03 for the envelope, and 0.04 for the spectrogram, but also presented results for independent frequency bands, obtaining 0.08 and 0.09 respectively for the Theta band. However, they decided to use the broad-band, given that relative differences between features were preserved using separate bands or a broad-band.

In the published version of the Desai et al. (2021) paper, the correlation values presented in the figures were normalised to the maximum possible correlation from the data (and the figure changed), but the average uncorrected values were reported in the text. From there, the average uncorrected correlation values from the TIMIT model were 0.26 for the envelope feature, 0.31 for the pitch, 0.09 for the spectrogram, and 0.35 for what they called “Full Model”, that includes envelope, pitch and phonetic features. The performance for the spectrogram model in this case is strikingly low compared to the envelope performance considering the precedent of Di Liberto et al. 2015 where the spectrogram model presented a higher predictive power than the envelope. Our average correlation values were 0.26 for the envelope, and 0.37 for the spectrogram. This increase was in agreement with [DiLiberto2015]. The greater predictive power for the spectrogram feature was also expected considering the spectrogram carries more information about the stimulus than the envelope. The Movie Trailer model presented average values of 0.07 and 0.05 respectively [Desai2021]. However, given the nature of the stimuli that involves visual stimulation and sound effects, we think that the comparison with the TIMIT model is fairer. The differences in this case are that TIMIT has clean short pre-recorded phrases and we have unsubscripted dialogues. In that sense, the correlation values of our work are similar for the envelope feature to the ones in [Desai2021] and greater for the Spectrogram feature.

We acknowledge that our work shows significantly greater correlation values. The uncorrected correlation values of the envelope were similar to those in Desai et al (2021) [Desai2021preprint,Desai2021], but the ones for the Spectrogram were 4 times greater, than the spectrogram results for previous work, and 1.5 times greater than the envelope result for Desai et al (2021) [Desai2021preprint,Desai2021]. Also they followed the increased relationship that the Spectrogram presents over the Envelope correlations in [DiLiberto2015], but [Desai2021] showed a decrease in the spectrogram in relation to the envelope. First, it is worth mentioning that there are only a few papers on this topic at the moment and they don't fully agree. Both in the results, the methods, and the stimuli. Thus, more work is needed in order to fully uncover the impact of the different features and their interactions in the brain. We speculate that both the methods and the stimuli could explain these differences. Regarding the methods, for instance, using narrow frequency bands versus broader bands. And regarding the stimuli, for instance, using natural stimuli in which participants have to act in consequence (with certain content and timing), or the language that could favour one feature over the other.

Reference	Database	Input feature	Target	Uncorrected Correlations
Di Liberto et al. (2015)	Audiobook	Envelope	(1-15 Hz)	0.03 (1)
Di Liberto et al. (2015)	Audiobook	Spectrogram	(1-15 Hz)	0.04 (1)
Di Liberto et al. (2015)	Audiobook	Envelope	Theta	0.08 (2)
Di Liberto et al. (2015)	Audiobook	Spectrogram	Theta	0.09 (2)
Desai et al. (2021)	TIMIT	Envelope	(1-15 Hz)	0.26 (3)
Desai et al. (2021)	TIMIT	Spectrogram	(1-15 Hz)	0.09 (3)
Ours	Natural Dialogue	Envelope	Theta	0.26
Ours	Natural Dialogue	Spectrogram	Theta	0.37
Ours	Natural Dialogue	Spectrogram	(1-15 Hz)	0.29

Table R1. Comparison between uncorrected correlations in different studies

(1) See Figure 2 [DiLiberto2015]

(2) See Figure 3 [DiLiberto2015]

(3) See Page 8952 [Desai2021]

3. Line 310: So, does that mean that you re-filter data that was filtered already. That is problematic. As you don't report the details of the second round of filters, it is impossible for me to estimate the impulse/step responses of the series of filters, which is worrying. Also, are we talking about zero-phase shift filters? Please clarify.

Response: We thank the reviewer for their timely comment regarding the details of the filtering process. As the reviewer correctly pointed out, we first apply a broad-band and a notch filter (before ICA; see page 18), and then filter again in narrower frequency bands. For this, we use a Minimum-phase lag causal filter (see page 19 and Supplementary Section F for further details).

Briefly, the reason why minimum phase filters were used is that non-causal zero-phase filters would modify the temporal causality in the EEG signal, which would have considerable and undesirable implications in the TRF fitting and the timing. According to [Etard2019] the TRF results from causal and non-causal filters only slightly, with a delay in the response of around 50 ms for causal filters. In our case, we had a 100 ms difference between the two filters, as it can be observed in the new Figure F.1 in the Supplementary section (also depicted below as R8). Moreover, the linear phase filter and the causal filter presented opposed polarisations in the mTRFs, where the causal filter showed results agreeing with the previous literature [Lalor2009, Lalor2010, DiLiberto2015].

Figure R8. Average mTRF of all participants fitted using spectrogram features as input in the Theta band from -300 ms to 400 ms as target. The top panel shows the mTRF when the EEG signal was filtered using a causal filter. The lower panels show the same procedure when using non-causal zero phase filters.

4. Was the data re-referenced to the mastoids (I assume so, but it's not indicated in the text)? The TRFs don't show much polarization (how does the GFP look like?). Is that all that response coming from the mastoids for some reasons? How do the results look like with a different referencing.

Response: The data was referenced to linked-mastoids. Our TRF are similar to those in DiLiberto et al. (2015), who also used mastoids as reference. Following the reviewer's suggestions, we ran a new analysis where the data was re-referenced to the EEG average. The procedure description and new results are now included in section G of the Supplementary information. The resulting TRF are similar to those presented in Etard et al. (2019), who also used average-reference, showing a high polarisation in the GFP peaks. Another observation is that the predictive power in central channels drops significantly, as they are predominant in the re-referencing to the average of all channels. Below, the mTRF obtained using the mastoid (top) and average (bottom) are depicted for an easier comparison for the reviewer.

Figure R9 mTRFs for the spectrogram in the Theta band averaged across participants. The top panel shows the mTRF for each electrode referenced to linked-mastoids, averaged across mel-bands (same as Figure 2.3A). The position of each electrode is indicated by the scalp plot on the top-left corner. The gray area on the bottom indicates the Global Field Power. The bottom panel shows the mTRF for each electrode re-referenced to the average, and averaged across mel-bands (same as Supplementary Figure G.1A). Scalp distributions at the time of the peak are presented on the top with the corresponding times.

Other comments

5. Abstract. It is unclear what the authors mean with “better performances” (Stronger effects of SIS?)

Response: We thank the reviewer for the comment. By better performance of the model, we mean greater correlation values obtained from the encoding model, compared with the ones obtained in other papers [Desai2021,DiLiberto2015]. In the present version of our manuscript we added this clarification in line 8.

6. Line 79: It's unclear if the authors intend to analyze perception only at first or production. I assume perception, since the text says that they are trying to replicate previous work initially

(but the text does not indicate which exact previous work we should refer to among the many ones cited above)

Response: As the reviewer correctly points out, our goal is to analyse only the perception of speech. The first part of our manuscript aims to validate the performance of the encoding model on our unscripted natural dialogues dataset.

As stated, we analysed the perception and aim to replicate diverse results from previous work. We replicate results regarding predictive power of the model for the Envelope and Spectrogram (and even Pitch) from [Desai2021,DiLiberto2015] (see Table R1). Also, regarding the regions of higher representation (ie. regions with greater correlation values) similar to those presented in Di Liberto et al (2015) were obtained. Moreover, the TRFs obtained in our work resemble those known in the literature [Lalor2009,Lalor2010,DiLiberto2015]. Finally, by analysing our data in frequency bands, we found analog results to the backward (decoding) model presented by Etard et al (2019), showing greater acoustic stimuli representation in Delta and Theta bands. We added more information in the Results section lines 95-98 to better clarify this point.

7. Line 60: True, but is imprecise. The parameters are called TRF because they estimate the TRF. But there are many ways to estimate a TRF. To be fair, even a cross-correlation is an estimate of the TRF (even if less accurate). This is me being picky here. This one is not important.

Response: Following this recommendation we further clarify this point in lines 62-66.

8. Fig. 2.2C. The ylabel is incorrect. I should be "Frequency (Hz)"

Response: We again greatly appreciate the reviewer's careful read of the manuscript. We correct this mistake in the new version.

9. Fig. 2.2B-Right. That is visually confusing and not convincing, honestly. But I won't go into detail until it's clear why those prediction correlation values are so high

Response: In Fig 2.2 B the correlation obtained using a left and right set of electrodes is depicted for the Envelope and Spectrogram features. Regarding the results for the envelope, there is no significant difference between the correlation values obtained from the right and left set of electrodes (Wilcoxon signed-rank test; Fig. 2.2B right panel). As pointed out before, in question #2, these values are in a similar order of magnitude that the ones obtained in previous related works [Desai2021]. Regarding the spectrogram, the correlation values obtained using the right set of electrodes are significantly higher than the ones using the left set (Wilcoxon signed-rank test; Fig. 2.2B left panel). In the new version of the manuscript, we updated the way of selecting the channels for the lateralization comparison effect, by now choosing the 12 channels in the frontal region with higher correlation values for each hemisphere. We also added a new section (C) in the Supplementary information with further details of the Lateralization analysis performed.

10. Line 111 and 312. Broad ERP band? Do you mean “EEG”? Or simply broader-band?

Response: We thank the reviewer for pointing this typo out. We corrected to broad-band (0.1-40 Hz) in the new version of the manuscript.

11. The caption of figure 2.2 could be clearer. It should stand on its own. Instead, it’s unclear what “frequency band” refers to (in this case, EEG frequency band) and what the correlation is calculated on.

Response: Following the reviewer’s suggestion, we split the previous Figure into the current Figures 2.2 and 2.3, for easier interpretation. Also, following the reviewer’s suggestion, we included further details in the caption of Fig 2.2 and 2.3 in the new version of the manuscript.

12. How were the electrodes selected for the lateralisation analysis

Response: Originally, the electrodes were selected by visual inspection, looking for the electrodes that matched the higher regions in both hemispheres. In the new version of the manuscript we updated this by selecting the electrodes separately for each hemisphere. We selected the 12 channels in the frontal region with higher correlation values for each hemisphere as explained in response 9. We include the details in the new Section C in the Supplementary information.

13. Line 314: It’s fine that you explain that you used MNE for the filtering. However, you should indicate the version of MNE, as well as all the details of the filters (not just what function you used).

Response: We agree those details were missing. Now, the details of the filtering and the packages and versions used are included in the new version of our manuscript in lines 363-373 and in Supplementary section F. We also would like to point out that all versions and environment used to obtain the results reported in this manuscript are detailed in the repository: <https://github.com/jegonza66/SIS-during-natural-dialogue/>

14. Line 309: What do you mean with “supervised by an expert”? Was that one of the authors? Could you not simply say “manually detected”? Also, are there some parameters that one should specify with those functions (as you did on line 303)? Keep in mind that this must be replicable.

Response: We thank the reviewer for pointing out the replication of the results. As correctly pointed out by the reviewer, the selection was supervised by one of the authors of the manuscript. We corrected the statement in the new version and expanded on the independent component selection criteria in the Section 4.3 (lines 348-362) (also in accordance with question #1).

15. Line 130: “An absence of response” s a tricky thing to say, statistically speaking. Maybe you mean that no TRF component was significantly above chance?

Response: We thank the reviewer for this important observation. We referred to the lack of significance in the correlation values. Now, we changed the statement for “no evidence of response” in the new version, as well as we added Cohen’s d-prime analysis to assess the effect size between the correlation value distributions in Standard deviation units.

16. Figure 2.3 should be improved from a graphical perspective (e.g., hard to read, labels not aligned).

Response: We improved the quality and size of the labels in the new version of the manuscript.

17. How is the “silence” TRF calculated? i.e., what is the stimulus vector in the encoding model?

Response: The stimulus vector in the encoding model is extracted from the other participant’s microphone in the time intervals when neither of the participants was speaking. The silence interval was determined by the manual annotations that are provided in the dataset [Gravano2023].

18. The TRFs look delayed compared to previous studies (using encoding models). The N1 usually emerges around 80ms, and the P2 around 140-160ms (in adults). However, the only discussion about that seems to be on lines 115 and 204, where the TRF is compared with ERPS instead.

Response: As stated in [Lalor2006,Lalor2010,Eingher2019,Crosse2016,Crosse2021], there is a direct relationship between the mTRF and the ERPs. Their major differences being the assumption of linear relationship between stimulus features and EEG signals, and isolating the response to specific features entangled within a stimulus. It is in this sense that we compare the obtained mTRFs to the evoked potentials from auditory stimuli.

Regarding the time delays, as previously stated in response 3 (and in the Methods in lines 363-373 and Supplementary Section F), in order to filter the signal in different frequency bands, linear phase filters were implemented to maintain the temporal causality of the signals. In a previous work [Etard2019], they compared the results from causal and non-causal filters, and obtained similar results but presenting a time delay of approximately 50 ms in the TRFs from causal filters, compared to non-causal.

We incorporated these comments in the Introduction and the Results sections (lines 126-133). As mentioned before, we include a Filtering and Pre-Stimulus Onset section in the Supplementary material section discussing the filtering impact.

19. Lines 186 and 205: That statement is incorrect. A few studies focused on the 1-15Hz band, but there is a lot of variability in the field in this regard. Also, please note that on of the two studies included as a reference to support that statement actually looked into the various sub-bands. Furthermore, saying that theta-band shows “greater representation” of acoustic

features s debatable. You could say that the prediction correlations are larger than in other bands. But that could mean many other more methodological things. Also, line 209: “main frequency band” and “this could be due to the cortical entrainment effect”. All these statements are vague and, frankly, it is unclear what the authors are trying to say.

Response: We thank the reviewer for their comments. From the bibliography we could find, the works with EEG that used encoding models for continuous audio representation were the ones cited [Lalor2009,Lalor2010,Desai2021,DiLiberto2015,Etard2019] and others that did not use speech [OSullivan2015] or envelope/spectrogram features [Khalighinejad2017]. As correctly pointed out, [DiLiberto2015] looked into different frequency bands, and obtained values 1.5 times greater for prediction correlations, when compared to the 1-15 Hz band, which concurs with our statement regarding the reason for a performance improvement. [Etard2019] also analyses the different frequency bands, but the performance in that work is computed for the prediction of a Decoding model, estimating the audio envelope from the EEG signal.

Following this reviewer's comments, we now replace the expression “higher representation” for “predictive power” as it was not what we aimed to state. In that regard, we also changed the term “Main frequency band”.

Regarding the cortical entrainment effect, we meant to sustent the observed higher correlations in the Theta band by recalling a well established effect of “cortical entrainment” [Giraud2012, Poeppel2020], considering that the frequency of syllable pronunciation in english is around 4-7 Hz [Ding2016], slightly below the syllable pronunciation rate in Spanish [Pellegrino et al., 2011]. We further discussed this matter in the Discussion section lines 245-255 to make it clearer.

20. Line 221. This is EEG. It is fine if the authors want to speculate on that, as long as it's clear that it is only a speculative reflection. However, I recommend not to go down that rabbit hole. This is only EEG!

Response: We agree with the reviewer, nevertheless within the discussion we tried to link our results with the existing bibliography. We rewrote the sentence in line 263-265.

21. Line 176. This paragraph is absolutely misleading. Those numbers are from different metrics. As far as I understand, some of that work uses a re-scaled prediction correlation (based on an ideal maximum estimated based on the trial-by-trial noise), while others do not. That discussion has to be corrected and the methods regarding this specific paper should also be much clearer about the procedure here.

Response: As correctly stated by the reviewer, [Desai2021] uses a re-scaled correlation, whereas the other work does not. However, by analysing the plots in [Desai2021] we believe that only the plots were corrected, and not the reported values in the text, as it doesn't seem that the average of those plots would yield such low values as 0.26 for example. Also, when referring to the figure, and in the figure caption, the authors always clearly stated “noise ceiling-corrected correlation values”, whereas they do not make such a statement when reporting the average values (“The average correlation value...”). In that sense, we believe that

the reported averaged values in the text correspond to the average of uncorrected correlation values, presented in the pre-print figure.

In our case, just like [DiLiberto2015,Hamilton2018,Hamilton2020], the metric is the Pearson correlation value, in our case averaged first across folds for every subject, then across subjects for every channel. Then, all the values reported in that paragraph correspond to Pearson correlation.

We better explained this comparison at the end of that paragraph in lines 218-221 in the new version of our manuscript.

22. Line 192: That also depends on how the envelope and spectrogram are calculated.

Response: Both [DiLiberto2015, Desai2021] computed the spectrogram using 16 mel-bands (as the envelope computed over the corresponding frequencies of each band), and 15 mel-bands respectively. There are no details in [Desai2021] about the computation of the spectrogram. In our work we used 16 mel-bands following [DiLiberto2015].

23. What happens for pre-stim lags? It is unclear why that is not included. This is particularly relevant as we are talking about speech production. Pre-stim responses would not only be related to muscle movement and motion! That would also be a further validation on the author's assumption that their denoising is actually removing the motion/motor components.

Response: We thank the reviewer for this remark, as it is very accurate. This was originally discussed only in the Supplementary Information (see supplementary section E). Indeed, the pre-stimulus times in the EEG show no response whatsoever, either to the external stimulus or to the self-speech (see Figure R10).

Figure R10. TRF of the spectrogram model in the Theta band with pre-stim times, for 2 conditions: Left: Only listening to external speech, Right: Listening to external speech while both are speaking.

For representation purposes, we have now updated all the corresponding figures in the text presenting the mTRFs, to include the pre-stim time lags. However, the predictions from the model were done using only positive times, to avoid providing the model with information from future time-points.

24. I might have missed it. Is there a data sharing statement?

Response: Speech data was part of the UBA Games Corpus, which was already released in an institutional public repository (<https://ri.conicet.gov.ar/handle/11336/191235>) [Gravano2023]. EEG data can be found in <https://figshare.com/s/53bcdc00470c59e29605> [private link, the permanent will be published upon acceptance] and the code to replicate the results obtained in this manuscript can be found in <https://github.com/jegonza66/SIS-during-natural-dialogue/>. We include this information in the new version of our manuscript.

25. Some details are missing in the caption for Figure 2.5.

Response: Following the reviewer's suggestion, we included more details in the caption of Figure 2.6 in the new version of the manuscript.

References:

- [Bigdely-Shamlo2013] Bigdely-Shamlo, N., Kreutz-Delgado, K., Kothe, C., & Makeig, S. (2013, July). EyeCatch: Data-mining over half a million EEG independent components to construct a fully-automated eye-component detector. In *2013 35th Annual International Conference of the IEEE Engineering in Medicine and Biology Society (EMBC)* (pp. 5845-5848). IEEE.
- [Crosse2016] Crosse MJ, Di Liberto GM, Bednar A, Lalor EC. The multivariate temporal response function 547 (mTRF) toolbox: a MATLAB toolbox for relating neural signals to continuous stimuli. *Frontiers in human neuroscience*. 2016;10:604.
- [Crosse2021] Crosse MJ, Zuk NJ, Di Liberto GM, Nidiffer A, Molholm S, Lalor EC. Linear Modeling of Neurophysiological Responses to Naturalistic Stimuli: Methodological Considerations for Applied Research. *551 PsyArXiv*; 202
- [Desai2021] Desai M, Holder J, Villarreal C, Clark N, Hoang B, Hamilton LS. Generalizable EEG encoding models with naturalistic audiovisual stimuli. *Journal of Neuroscience*. 2021;41(43):8946-62.
- [DiLiberto2015] Di Liberto GM, O'Sullivan JA, Lalor EC. Low-frequency cortical entrainment to speech reflects phoneme-level processing. *Current Biology*. 2015;25(19):2457-65.
- [Ding2016] Ding, N., Melloni, L., Zhang, H., Tian, X., & Poeppel, D. (2016). Cortical tracking of hierarchical linguistic structures in connected speech. *Nature neuroscience*, 19(1), 158-164.
- [Eingher2019] Ehinger, B. V., & Dimigen, O. (2019). Unfold: an integrated toolbox for overlap correction, non-linear modeling, and regression-based EEG analysis. *PeerJ*, 7, e7838.
- [Etard2019] Etard O, Reichenbach T. Neural speech tracking in the theta and in the delta frequency band differentially encode clarity and comprehension of speech in noise. *Journal of Neuroscience*. 2019;39(29):5750-9.
- [Giraud2012] Giraud AL, Poeppel D. Cortical oscillations and speech processing: emerging computational principles and operations. *Nature neuroscience*. 2012;15(4):511-7.
- [Hamilton2018] Hamilton, L. S., Edwards, E., & Chang, E. F. (2018). A spatial map of onset and sustained responses to speech in the human superior temporal gyrus. *Current Biology*, 28(12), 1860-1871.

- [Hamilton2020] Hamilton, L. S., Oganian, Y., & Chang, E. F. (2020). Topography of speech-related acoustic and phonological feature encoding throughout the human core and parabelt auditory cortex. *BioRxiv*, 2020-06.
- [Lalor2006] Lalor, E. C., Pearlmuter, B. A., Reilly, R. B., McDarby, G., & Foxe, J. J. (2006). The VESPA: a method for the rapid estimation of a visual evoked potential. *Neuroimage*, 32(4), 1549-1561.
- [Lalor2009] Lalor EC, Power AJ, Reilly RB, Foxe JJ. Resolving precise temporal processing properties of the auditory system using continuous stimuli. *Journal of neurophysiology*. 2009;102(1):349-59.
- [Lalor2010] Lalor EC, Foxe JJ. Neural responses to uninterrupted natural speech can be extracted with precise temporal resolution. *European journal of neuroscience*. 2010;31(1):189-93.
- [Perez2022] Pérez, A., Davis, M. H., Ince, R. A., Zhang, H., Fu, Z., Lamarca, M., ... & Monahan, P. J. (2022). Timing of brain entrainment to the speech envelope during speaking, listening and self-listening. *Cognition*, 224, 105051.
- [Poeppel2020] Poeppel D, Assaneo MF. Speech rhythms and their neural foundations. *Nature reviews neuroscience*. 2020;21(6):322-34.
- [Mognon2011] Mognon, A., Jovicich, J., Bruzzone, L., & Buiatti, M. (2011). ADJUST: An automatic EEG artifact detector based on the joint use of spatial and temporal features. *Psychophysiology*, 48(2), 229-240.

New references:

- [Chartier2018] Chartier, J., Anumanchipalli, G. K., Johnson, K. & Chang, E. F. (2018) Encoding of articulatory kinematic trajectories in human speech sensorimotor cortex. *Neuron* 98((5) jun), 1042–1054.e4
- [Desai2021preprint] Maansi Desai, Jade Holder, Cassandra Villarreal, Nat Clark, Liberty S. Hamilton “Generalizable EEG encoding models with naturalistic audiovisual stimuli” bioRxiv 2021.01.15.426856; doi: <https://doi.org/10.1101/2021.01.15.426856>
<https://www.biorxiv.org/content/10.1101/2021.01.15.426856v1>
- [Gravano2023] Gravano, Agustin; Kamienkowski, Juan Esteban; Brusco, Pablo; (2023): UBA Games Corpus. Consejo Nacional de Investigaciones Científicas y Técnicas. (dataset). <http://hdl.handle.net/11336/191235>
- [Jansen2020] Janssen, N., Meij, M. V. D., López-Pérez, P. J., & Barber, H. A. (2020). Exploring the temporal dynamics of speech production with EEG and group ICA. *Scientific reports*, 10(1), 3667.
- [Khalighinejad2017] Khalighinejad, B., da Silva, G. C., & Mesgarani, N. (2017). Dynamic encoding of acoustic features in neural responses to continuous speech. *Journal of Neuroscience*, 37(8), 2176-2185.
- [OSullivan2015] O'Sullivan, J. A., Shamma, S. A., & Lalor, E. C. (2015). Evidence for neural computations of temporal coherence in an auditory scene and their enhancement during active listening. *Journal of Neuroscience*, 35(18), 7256-7263.
- [Pellegriano2011] Pellegriano, F., Coupé, C., & Marsico, E. (2011). A cross-language perspective on speech information rate. *Language*, 539-558.

- [Porcaro2015] Porcaro, C., Medaglia, M. T., & Krott, A. (2015). Removing speech artifacts from electroencephalographic recordings during overt picture naming. *NeuroImage*, 105, 171-180.
- [Muthukumaraswamy2013] Muthukumaraswamy, S. D. (2013). High-frequency brain activity and muscle artifacts in MEG/EEG: a review and recommendations. *Frontiers in human neuroscience*, 7, 138.
- [Tran2004] Tran, Y., Craig, A., Boord, P., & Craig, D. (2004). Using independent component analysis to remove artifact from electroencephalographic measured during stuttered speech. *Medical and biological engineering and computing*, 42, 627-633.

Reviewers' comments:

Reviewer #1 (Remarks to the Author):

The authors have provided acceptable answer to my suggestions and concerns.

Things that I still find a problematic and which the authors may want to consider, for sake of their own reputation and the benefit of the readers:

Statistical reporting requires the degrees of freedom to each and every test. This is still not in the manuscript.

There are still several suggestions for a lack of an effect, which minor word-smithing did not change.

Reporting Bayes Factors is really not hard to do. I have found this code easy-to-use:

<https://klabhub.github.io/bayesFactor/> It can compute the BF along with the p-values for most conventional statistical tests.

Extensive explanations are provided to reviewers in the response document, but appear to be missing in the manuscript. I suspect readers would appreciate many of these clarifications as well. Consider including some of them.

Reviewer #3 (Remarks to the Author):

Title: "Speech-induced suppression during natural dialogues"

I appreciate the authors effort in addressing my previous comments. While some issues were addressed, the remains fundamental problems that, in my opinion, mine the validity of the analysis. I'll include the main issues below.

Major comments

1. Filtering. I am afraid that the authors' answer reinforces my concerns. I had previously highlighted the risks that come with filtering.

a. First, regarding the causal vs. non-causal filter. The authors show that the two filters lead to completely different TRFs. That is worrying and only highlights that the result seems to be largely dependent on the choice of filters. The authors show more than a simple shift, but an actual change in the polarity of the TRF. Again, that is worrying. Using a zero-phase shift filter (e.g., by using the same filter twice in opposite directions, with the function `filtfilt`) is typically done in the literature to preserve the latency. Since the consequence is that the TRF components will be "larger" (so, they may start rising before that is actually "true" in the neural signal), only the peak of the TRF components is usually studied, as the timing of the peak is unaffected. Instead, using a causal filter changes that time, which is a problem.

b. Some details of the filter are still missing. Nonetheless, I tried replicating some of the filters in the

study. The first high-pass filter, for example, with an order of 16896, seem to correspond to worryingly delayed impulse responses. Could the authors provide the impulse responses of their exact filters (as well as complementing the description of the filters in the manuscript – not that mentioning the tool/function used is insufficient – one should be able to replicate the filter with whatever toolbox).

c. In my understanding, the authors insisted with refiltering the data multiple times, rather than fixing the issue. Again, that is worrying, as the negative impacts of the filters will accumulate, as I explained in the previous review. The authors should report the impulse response of the individual filters and of the combined filters.

d. In sum, it is possible that the results are largely reflecting the filter used (as showed by the authors themselves in the rebuttal letter). As such, in addition to fixing the issue and reporting the information mentioned above, I suggest running the analysis with milder filters of different type (e.g., Butterworth), as done in recent TRF studies.

2. The authors have not convinced be on the point of the prediction correlation values. EEG signals are mostly noise, so those numbers are not believable (I am referring to the uncorrected ones). I suggest analysing the listening portion of the data as in previous studies to see if the difference is really due to the stimuli (as claimed by the authors) or to the filters and ICA used. So, running the analysis without ICA and with milder zero-phase shift filters, to rule out the likely possibility that the current results are driven by filtering artifacts.

3. I have similar concerns regarding the ICA, but that is harder for me to evaluate without seeing the components removed and the resulting EEG data.

Overall, I don't find the results convincing, and instead they appear to be the result of methodological flaws. As always in science, I might be wrong. But this is my honest opinion on the current version of this manuscript.

Reviewer #1 (Remarks to the Author):

Comment 1: The authors have provided acceptable answer to my suggestions and concerns.

Things that I still find a problematic and which the authors may want to consider, for sake of their own reputation and the benefit of the readers:

Response 1: Thanks for the positive comments, we address the remaining issues below,

Comment 2: Statistical reporting requires the degrees of freedom to each and every test. This is still not in the manuscript.

Response 2: Thanks to the reviewer's observation, we now included the degrees of freedom (d.f.) in the Methods section and every statistical statement on pages 7, 8, and 11. Briefly, for the comparison of the correlation values between conditions (Fig. 2.5A) the d.f. is also equal to 17 (considering 18 participants), for each comparison (before multiple comparisons correction) in the TFCE approach (Fig. 2.3B) the d.f. is equal to 17 (considering 18 participants), and for the lateralization analysis (Fig. 2.2B) the d.f. is equal to 11 (considering 12 channels). Finally, for the permutation test to assess the model significance it is not correct to report a d.f. because we are not using any defined null distribution, instead we reported the number of observations (when mentioned) and the number of permutations (in Methods section).

Comment 3: There are still several suggestions for a lack of an effect, which minor word-smithing did not change. Reporting Bayes Factors is really not hard to do. I have found this code easy-to-use: <https://klabhub.github.io/bayesFactor/> It can compute the BF along with the p-values for most conventional statistical tests.

Response 3: We appreciate the reviewer's feedback and the repository link. We include the Bayes Factor analysis [Kass and Raftery, 1995; Hoijtink, et al., 2019] in (Figure 2.5) and discuss pages 9 and 10. From this analysis, we did not find a conclusive result that can support the complete inhibition of the response of one's own speech. However, there is a lot of convergent evidence and at least a strong attenuation. We changed the manuscript in accordance.

In Figure R1 we contrast the evidence in favor of the alternative hypothesis (H1), which is that there is an effect (a difference between conditions), and in favor of the null hypothesis (H0), which is the absence of the effect. The evidence in favor of one hypothesis over the other is estimated by the Bayes Factors (BF10 and BF01 respectively). In the first two rows of each panel, we can observe that there was a larger effect of the Listening condition over the others. Including a larger response of the Listening alone versus Listening while both are speaking that could be explained by the larger number of samples (see Supp. Fig. H.1). These results are consistent with the results already presented in the previous version of the manuscript.

In the second two rows we compare the Speaking conditions (Speaking alone: (S) and Speaking while both are speaking (S|B)) between them and with the Silence condition. When comparing (S) versus Silence we observed that all channels favor the H0, and there is substantial evidence in 58 channels. In the case of (S|B) versus (S) or Silence we find

mainly channels in which the evidence favors H0 but also channels in which the evidence favors H1. These last channels are mainly located in the fronto-temporal regions and could be due to the presence of signals of the speech of the other participant in their own channel.

Overall, these results support previous findings and stress the evidence in favor of the lack of response in the Speaking condition. Nevertheless, as the magnitude of the BF01s is small and there is also some evidence in favor of H1, the results are not conclusive and we will replace the “lack of response” with a “very strong attenuation of the response” that could be supported by different approaches (TRF and PLV) and different visualizations (TRF and correlations) and statistics (permutation test of each condition and signed-rank, Cohen’s d prime, and Bayes Factors between conditions).

References:

[1] Kass, R. E. & Raftery, A. E. Bayes factors. *J. Am. Stat. Soc.* 90, 733-795 (1995). <https://doi.org/10.1080/01621459.1995.10476572>

[2] Hoijtink, H., Mulder, J., van Lissa, C., & Gu, X. (2019). A tutorial on testing hypotheses using the Bayes factor. *Psychol. Meth.*, 24(5), 539–556. <https://doi.org/10.1037/met0000201>

Comment 4: Extensive explanations are provided to reviewers in the response document, but appear to be missing in the manuscript. I suspect readers would appreciate many of these clarifications as well. Consider including some of them.

Response 4: Thanks for the careful reading, we tried to balance the information and readability, and we included many aspects of the reviewer’s response to the supplementary material. Now, we revised the manuscript trying to include some comments and references that we had left out before. Firstly, we expand the explanation on how the correlation values were summarized across participants and electrodes (see *Methods>Model Performance*)

that completes the explanation on how they were calculated and the discussion on how other researchers estimated them (and why that does not apply here). Secondly, we now included a figure showing the mTFR to the 1-8 Hz band with a discussion about the filters implemented, that expands the response on analyzing different frequency bands and Theta in particular. Thirdly, in the current version, we included the condensed explanation on the model and the permutation in Supplementary Section I with the Figure that we prepared for our previous responses, accompanying the explanations on *Methods > Encoding models* and *Methods > Model's performance*. Fourthly, we rewrote the Results section to include the Bayes Factors analysis, with its corresponding explanation in *Methods > Models' comparison Methods subsection*, and we added a comment on these results in the Discussion section. Finally, we added a new supplementary material on different filters and a subsection in the Discussion with a presentation of methodological decisions and their implications (ICA, filters, and how the correlation values are calculated).

Reviewer #2 (Remarks to the Author):

Comment 1: Title: "Speech-induced suppression during natural dialogues"

I appreciate the authors effort in addressing my previous comments. While some issues were addressed, the remains fundamental problems that, in my opinion, mine the validity of the analysis. I'll include the main issues below.

Major comments

1. Filtering. I am afraid that the authors' answer reinforces my concerns. I had previously highlighted the risks that come with filtering.

a. First, regarding the causal vs. non-causal filter. The authors show that the two filters lead to completely different TRFs. That is worrying and only highlights that the result seems to be largely dependent on the choice of filters. The authors show more than a simple shift, but an actual change in the polarity of the TRF. Again, that is worrying. Using a zero-phase shift filter (e.g., by using the same filter twice in opposite directions, with the function `filtfilt`) is typically done in the literature to preserve the latency. Since the consequence is that the TRF components will be "larger" (so, they may start rising before that is actually "true" in the neural signal), only the peak of the TRF components is usually studied, as the timing of the peak is unaffected. Instead, using a causal filter changes that time, which is a problem.

b. Some details of the filter are still missing. Nonetheless, I tried replicating some of the filters in the study. The first high-pass filter, for example, with an order of 16896, seem to correspond to worryingly delayed impulse responses. Could the authors provide the impulse responses of their exact filters (as well as complementing the description of the filters in the manuscript – not that mentioning the tool/function used is insufficient – one should be able to replicate the filter with whatever toolbox).

c. In my understanding, the authors insisted with refiltering the data multiple times, rather than fixing the issue. Again, that is worrying, as the negative impacts of the filters will accumulate, as I explained in the previous review. The authors should report the impulse response of the individual filters and of the combined filters.

d. In sum, it is possible that the results are largely reflecting the filter used (as showed by the authors themselves in the rebuttal letter). As such, in addition to fixing the issue and reporting the information mentioned above, I suggest running the analysis with milder filters of different type (e.g., Butterworth), as done in recent TRF studies.

Response 1: We thank the reviewer for their careful reading, please find point-by-point responses below. We hope to dispel the reviewer's doubts.

First of all, all the filters are applied to the continuous data, and the conditions correspond to small cropped partitions of this data. Thus, alterations in the signal of interest and the correlations should be observed in all the conditions. Then, in Supp Fig F1 (causal vs non-causal) we showed a displacement of the signal, but in both situations the response is clear and the effects are present.

(a. / d.) The reviewer says that the displacement of the signal of interest when comparing causal and non-causal filters is "... worrying and only highlights that the result seems to be largely dependent on the choice of filters" As it can be seen in the figures below, regardless of the filters applied, the correlation values (see Table R2) as well as the scalp distribution of the TRF are the same. Only the latencies of the TRF peaks change. In the manuscript, we rephrased the statements on the latencies, the only important message is that there is no signal observable before zero and that the most relevant information is in the first 200 ms. We think that the exact latencies are not really relevant to our results, and the take-home message is that when applying non-causal filters the audio samples from the future are not used for the prediction of the EEG sample at time zero. But most importantly, The statistical tests that endorse our main results are based on the correlation values obtained from the model (not the TRFs), and as it can be seen in the figure below, those values are robust to the different filtering methods proposed by the reviewer.

Filter	Frequency band	Condition	Correlation
FIR	Theta	All listening: '(E)'	0.367 ± 0.032
FIR	Theta	All speaking: '(S)'	0.020 ± 0.005
Butterworth	Theta	All listening: '(E) + (E B) + Silence'	0.333 ± 0.029
Butterworth	Theta	All speaking: '(S) + (S B) + Silence'	0.014 ± 0.005
FIR	Theta	All listening: '(E) + (E B) + Silence'	0.308 ± 0.029
FIR	Theta	All speaking: '(S) + (S B) + Silence'	0.013 ± 0.005
Butterworth	1-8Hz	All listening: '(E) + (E B) + Silence'	0.301 ± 0.027
Butterworth	1-8Hz	All speaking: '(S) + (S B) + Silence'	0.022 ± 0.005
FIR	1-8Hz	All listening: '(E) + (E B) + Silence'	0.276 ± 0.028
FIR	1-8Hz	All speaking: '(S) + (S B) + Silence'	0.040 ± 0.009

Table R1: Correlations values for the different combinations of filters and frequency bands. The FIR filter corresponds to the causal finite-impulse response filter implemented in our work, and described in the manuscript. The Butterworth filter is a non-causal 3rd order Butterworth filter suggested by the reviewer.

Then the reviewer states that “Since the consequence is that the TRF components will be “larger” (so, they may start rising before that is actually “true” in the neural signal), only the peak of the TRF components is usually studied, as the timing of the peak is unaffected”. As mentioned before, we are not interested in the exact timing (other than checking that the information from the future is not relevant). But also, we are in agreement with Etard et al. (2019) (see Figure 7 in Etard et al., 2019) where it is possible to see how the timing of the TRFs from causal filters is affected by a delay of approximately 50 ms, and not a broadening of the TRF, as stated by the reviewer.

(a. / b. / c.) The initial filters are broad enough (0.1 and 100 Hz) and applied to continuous data to prevent significant alterations in the ERPs. The successive filtering of causal filters is actually recommended rather than accumulating negative impacts as stated, for instance, in the EEGLAB web page (https://eeglab.org/others/Firfilt_FAQ.html). To apply a band pass filter, it is recommended to first apply a high pass filter and then a low pass filter. The filters applied in the preprocessing instance were all zero-phase non-causal filters, applied following the instructions of the EEGLab toolbox (Figure R3 presents the impulse response of both filters and the combination). Unfortunately, we do not have the raw data to undo the initial [0.1 100]Hz filters, but considering how broad the filters were and consistent with the recommendations from the EEGLab toolbox among others, we have no reason to doubt our results.

Figure R3: Impulse response of the initial broad filters.

The impulse response of the causal filter (in the Theta band) is presented in Figure R4A, but it was not combined with the impulse response of the previous filters because those were non-causal filters applied in both directions whereas this is a causal filter applied forward. To address the possibility of a “worryingly delayed impulse response” being introduced by the causal filters, we replicated the main analysis of the manuscript but in this case, using time windows centered in the past and in the future. We used non-overlapping time windows of 0.6s wide centered from -3.6s to 3.6s. A clear peak in the correlation is observed in the window centered around zero and, being a causal filter, it is reasonable to also observe an increase in the correlation before time zero as some of the past could influence the prediction (Figure R4B). No sign of any large introduced delay was observed.

Figure R4: A) Impulse response of the causal filter in Theta band. B) Mean correlation values and standard deviation using the spectrogram features as input and filtering in the Theta band with causal filters.

Comment 2: 2. The authors have not convinced be on the point of the prediction correlation values. EEG signals are mostly noise, so those numbers are not believable (I am referring to the uncorrected ones). I suggest analysing the listening portion of the data as in previous studies to see if the different is really due to the stimuli (as claimed by the authors) or to the filters and ICA used. So, running the analysis without ICA and with milder zero-phase shift filters, to rule out the likely possibility that the current results are driven by filtering artifacts.

Response 2: When revising previous studies we find similar correlation values for instance in [Desai 2021 preprint] and [Desai 2022] (please see the uncorrected correlation values mentioned in the text). Due to the nature of our stimuli, natural dialogues, we cannot run multiple repetitions of the same stimuli as in passive listening, which are necessary to estimate the ceiling corrected correlation values, as discussed in the Discussion section.

We are not sure what the reviewer refers to when suggesting "... analysing the listening portion of the data as in previous studies to see if the different is really due to the stimuli (as claimed by the authors)" because that is exactly what we do in Figures 2.3, 2.4 A, 2.6 A and then compare with the portions of the data in which the other was speaking, both are speaking, or both are silent in Figures 2.5. Nevertheless, showing larger correlation values in natural dialogues in comparison to passive listening is not the point of this study and we agree with the reviewer that we can not demonstrate here that difference. We are running a different set of experiments to address that question.

Finally, as we mentioned before, unfortunately, we do not have the raw data to undo the ICA step and compare. Nevertheless, in the manuscript, we argue why we think it could not produce the pattern of responses presented here. Please, find a discussion on the filters in the previous response.

Comment 3: 3. I have similar concerns regarding the ICA, but that is harder for me to evaluate without seeing the components removed and the resulting EEG data.

Response 3: ICA is a common approach for removing eye movements and noise from the EEG signal, please, find a detailed explanation of how it was done in the previous response and in the current version of the manuscript (section 4.3). Now, the reviewer can also find some examples of the scalp distributions below.

Horizontal EM	Vertical EM	Blinks	Discontinuities	"Jaws"
				s24-1	s24-1	s24-1	s24-1	s24-1
IC: 8	IC: 88	IC: 2	IC: 22	IC: 86

Horizontal EM	Vertical EM	Blinks	Discontinuities	“Jaws”
				s30-2	s30-2	s30-2	s30-2	s30-2
IC: 30	IC: 9	IC: 7	IC: 36	IC: 64

Moreover, it is clear that any correlation with muscular artifacts would have an immediate impact on the EEG signal, resulting in responses around $t=0$, but not in later times, where we are focusing our analysis. Also, the response of the EEG to auditory stimuli does not necessarily correlate with the muscular artifacts of the production. In that sense, the response to auditory stimuli would not match the muscular artifacts neither in time nor in shape, so we don't see the problem with this methodology.

We are not sure how the reviewer expects that removing ICs with that procedure affects the results in line with our observations. It would be great to have some references to follow.

Comment 4: Overall, I don't find the results convincing, and instead they appear to be the result of methodological flaws. As always in science, I might be wrong. But this is my honest opinion on the current version of this manuscript.

Response 4: We appreciate the honest concerns of the reviewer, and we learned a lot by responding to them. We do our best to respond to many of them during the iterations and include their concerns in the paper for the readers to judge although we do not agree with many of them. Nevertheless, we sincerely think there is enough evidence to support our claims.

REVIEWERS' COMMENTS:

Reviewer #3 (Remarks to the Author):

I appreciate the direct answer. The reply clarified some of my concerns, such as the little interest in the specific latencies, in this paper (but note that the readers might use that information in future work nonetheless). I remain doubtful on some points, such as whether correlation values around 0.4-0.5 are meaningful. I appreciate all the work done by the authors to address my comments. At this stage, we agree to disagree on such points, which is fine. I'll leave this to the editor and, in case of publication, to the readers. Again, I think the authors did a great job with this review process, even though not all comments could be addressed.